



# Seasonal dispersal of fjord meltwaters as an important source of iron to coastal Antarctic phytoplankton

Kiefer Forsch[1], Lisa Hahn-Woernle[2], Robert Sherrell[3], Joe Roccanova[3], Kaixan Bu[3], David Burdige[4], Maria Vernet[1], Katherine A. Barbeau[1]

[1]Scripps Institution of Oceanography, University of California San Diego, La Jolla, 92037, USA
[2]Department of Oceanography, University of Hawai`i, Manoa, 96822, USA
[3]Department of Marine and Coastal Sciences, Rutgers University, New Brunswick, 08901, USA
[4]Department of Ocean & Earth Sciences, Old Dominion University, Norfolk, 23529, USA

*Correspondence to*: Kiefer O. Forsch (kforsch@ucsd.edu)

**Abstract.** Glacial meltwater from the western Antarctic Ice Sheet is hypothesized to be an important source of cryospheric iron, fertilizing the Southern Ocean, yet its trace metal composition and factors which control its dispersal remain poorly constrained. Here we characterize meltwater iron sources in a heavily glaciated western Antarctic Peninsula (WAP) fjord. Using dissolved and particulate ratios of manganese-to-iron in meltwaters, porewaters, and seawater, we show that glacial melt and subglacial plumes contribute to the seasonal cycle of bioavailable iron within a fjord still relatively unaffected by climate change-induced glacial retreat. Organic ligands derived from the phytoplankton bloom and the glaciers bind dissolved iron and facilitate the solubilization of particulate iron downstream. Using a numerical model, we show that plumes generated by outflow from the subglacial hydrologic system, enriched in labile particulate trace metals derived from a chemically-modified crustal source, can supply the surface through vertical mixing, and that prolonged katabatic wind events enhance export of meltwater out of the fjord. Thus, we identify an important atmosphere-ice-ocean coupling intimately tied to coastal iron biogeochemistry and primary productivity along the WAP.

## 1 Introduction

Warm temperatures are accelerating glacial retreat and increasing meltwater discharge, rapidly changing Earth's cryosphere (Mouginot et al., 2019; Rignot et al., 2013). Ranging from diffuse flows to waterfalls and streams, cryospheric meltwaters deliver dissolved and particulate material, altering coastal ocean biogeochemistry. Glacial meltwater enters the ocean through surface runoff, direct melting of glacial ice (including icebergs), and discharge from liquid water reservoirs beneath glaciers, carrying iron (Fe) and other trace metals weathered from continental crust. In the surface ocean, the delivery of new Fe is critical for the growth of phytoplankton; and when enhanced, naturally or artificially, carbon is sequestered by the biological pump (Boyd *et al.*, 2019). However, direct measurements of Fe in heavily glaciated fjords reveal that up to 90-99% of dissolved Fe (dFe) originating from glaciers is removed upon mixing with seawater due to estuarine-type removal processes, including: precipitation of insoluble oxyhydroxides, adsorption to the surfaces of existing particles, and





aggregation of colloids and particles (Boyle et al., 1977; Schroth et al., 2014). Together, these processes are known as scavenging and constitute a major control on the distribution of Fe in the ocean by converting soluble forms of Fe into colloidal aggregates and particles (Wu *et al.* 2001). Constraints on the flux of newly delivered glacial Fe that escapes this

sink and is transported across continental shelves will enable better predictions of open ocean primary production and carbon sequestration, especially in oceanic regimes where Fe is a limiting nutrient.

Evidence for Fe delivery from the cryosphere is historically based on geochemical analysis of endmember glacial discharge (Hawkings *et al.,* 2014; Raiswell and Canfield 2012; Hodson *et al.,* 2017; Hawkings *et al.,* 2020), and discrete sampling of glacial ice (e.g. Hopwood *et al.* 2018) and seawater adjacent to marine-terminating glaciers and ice sheets (Hopwood *et al.,*

2016; Annett *et al.,* 2015; Gerringa *et al.,* 2015; Alderkamp *et al.,* 2012; Sherrell *et al.,* 2018). Trace metal studies at the ice-ocean interface have previously been conducted in fjords experiencing intense seasonal melt, such as in Alaska, Greenland, and Svalbard (Hopwood et al., 2016; Kanna et al., 2020; Schroth et al., 2014; Zhang et al., 2015). These temperate and high Arctic coastal waters experience large freshwater and sediment fluxes as a result of increased glacial discharge, which in turn create extreme physical and geochemical gradients. Ultimately, such dramatic changes in turbidity disturb local primary

production by decreasing light availability within the water column and reduced macronutrient supply (Meire et al., 2017). Even with high particulate and dissolved Fe contents, meltwaters from these fjords do not feed directly into offshore waters without significant scavenging, mixing and dilution (Hopwood *et al.*, 2015), bringing into question the effectiveness with which glacial meltwater-derived Fe may fertilize the surrounding ocean.

In Antarctica, fjords are less well-studied than their Arctic counterparts, but are also locations of intense seasonal blooms

with comparable or higher primary production relative to the adjacent continental shelves, and high sequestration efficiencies of organic carbon (Vernet *et al.*, 2008; Grange and Smith 2013; Taylor, DeMaster, and Burdige 2020). Along the western Antarctic Peninsula (WAP), 674 marine-terminating glaciers drain into the coastal ocean, primarily in fjords (Cook et al., 2016). The vast majority of these marine-terminating glaciers are retreating due to intrusions of warm deep water from the shelf, but many still remain cold-based (that is, local ocean temperatures are too cold to melt the glacier terminus),

particularly in the northern WAP where Weddell Water from the eastern side mixes with the Bransfield Strait (Cook et al., 2016; Pritchard and Vaughan, 2007). These glaciers are thought to have relatively small subglacial meltwater discharge, with suspended sediment signatures that spread laterally in the coastal ocean (Domack and Ishman, 1993; Domack and Williams, 2011). This makes cold glacio-marine Antarctic fjords unique locations for sampling subglacial discharge with minimal alteration.

Subglacial environments distinguish themselves from other cryospheric sources of Fe to the oceanic euphotic zone. Within the subglacial cavity, anoxia develops due to enhanced microbial respiration processes, high weathering rates, and limited diffusion of oxygen and exchange with the coastal ocean (Mikucki et al., 2009). The result is increased solubility of iron as Fe(II), and other redox sensitive elements, such as manganese (Mn). Meltwater discharge from beneath marine-terminating glaciers enters the ocean in the subsurface but may be mixed into the surface due to its positive buoyancy relative to

seawater. Enhanced vertical shear occurs episodically in the Antarctic coastal ocean as cooled dense parcels of air accelerate



down ice sheets, generating the strongest coastal winds on Earth (>30 m s⁻¹), near the coast. These episodic katabatic wind events could also be important for enhancing the supply of subsurface meltwaters to the surface (Jackson et al., 2014; Lundesgaard et al., 2019). The subglacial meltwater source represents a large uncertainty in the supply of cryospheric Fe to the ocean given the challenge of acquiring samples of the subglacial hydrologic system directly or with minimal alteration,

particularly in Arctic environments with intense seasonal melt flows and associated sediment turbidity (Straneo and Cenedese, 2015).

We present trace metal results from two expeditions (December 2015 and April 2016) to Andvord Bay, a cold glacio-marine fjord located mid-latitude along the WAP. This study is part of the FjordEco project which assessed the ecosystem function and seasonality of Andvord Bay (Pan *et al.*, 2019; Pan *et al.*, 2020; Ziegler *et al.*, 2020; Eidam *et al.*, 2019; Lundesgaard *et*

*al.*, 2020, 2019; Hahn-Woernle *et al.*, 2020). The WAP is host to the most extensive collection of glaciomarine fjords on the Antarctic continent, and its shelf waters are subject to ongoing biogeochemical and ecological alteration linked to large-scale changes to the western Antarctic Ice Sheet (Henley et al., 2020). We present a detailed picture of fjord Fe biogeochemistry and seasonality prior to significant glacier retreat, for which few ocean measurements exist.

## 2 Methods

### 2.1 Oceanographic setting and sampling

Andvord Bay is a glacio-marine fjord located mid-latitude along the west Antarctic Peninsula (WAP). This site was chosen because it has been identified as a productivity "hotspot" (Grange and Smith, 2013), and because of its proximity to long-standing ecological research programs (Palmer Long Term Ecological Research program). This location is characterized by converging deep water masses with distinct physical properties (relatively warm modified Upper Circumpolar Deep Water,

cold Weddell Water**)**. Bordering Andvord Bay are 11 marine-terminating glaciers (Fig. 1) with Moser and Bagshawe glaciers responsible for the majority of the solid ice flux. These glaciers are cold-based (-1 to -0.5 °C) resulting in weak meltwater signatures within the fjord (Lundesgaard *et al.*, 2020). Observations and sampling of Andvord Bay were conducted during two cruises as part of the FjordEco program during two cruises: LMG15-10 from 27 November to 22 December 2015 (late Spring) on R/V *Laurence M. Gould*, NBP16-03 from 4 April to 26 April (Fall) aboard R/V *Nathaniel B. Palmer*. On

December 11, 2015 a strong katabatic wind event, with peak along-fjord velocities of 30 m s⁻¹, was observed and lasted for 5 days. Atmospheric observations by two automatic weather stations (Neko Harbor, Useful Island) recorded episodes of high velocity katabatic winds between field seasons, showing that these are common events in this study region.



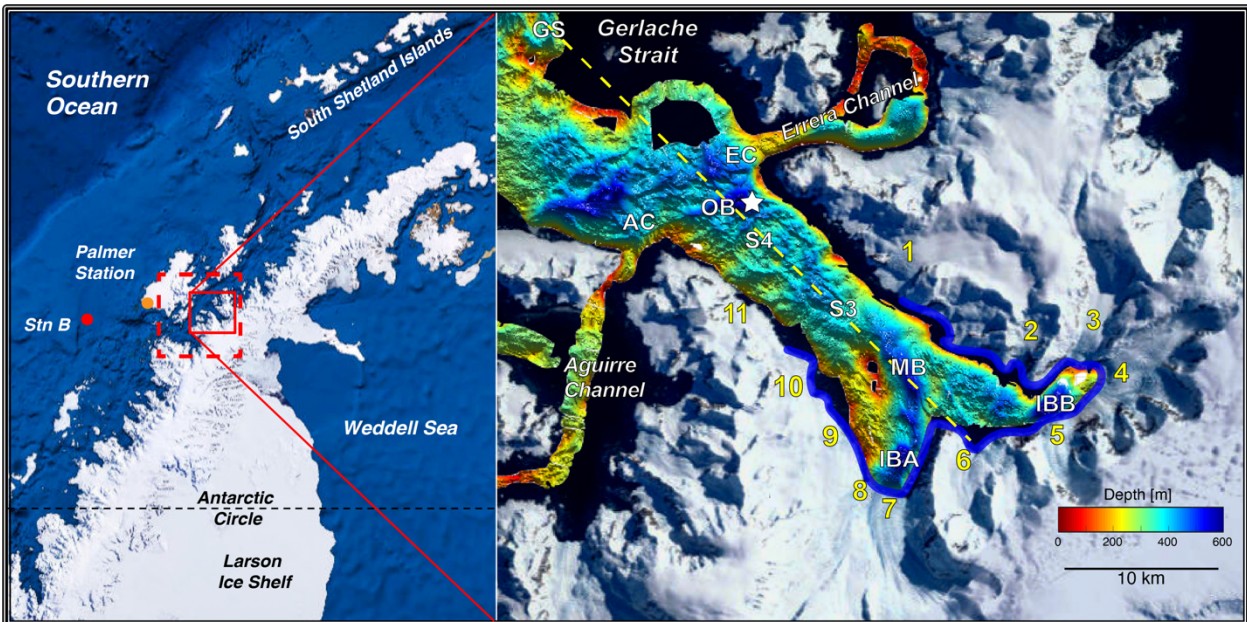

**Figure 1. Regional map of study region (red box, inset right) and model domain (dashed red box) with nearby Palmer Station and shelf station (Stn B). Bathymetric map of Andvord Bay with important stations labeled (GS = Gerlache Strait, AC = Aguirre Channel, EC = Errera Channel, OB = Outer Basin, S4 = Sill 4, S3 = Sill 3, MB = Middle Basin, IBA = Inner Basin A, IBB = Inner Basin B) and the surrounding tidewater glaciers numbered (4 = Moser Glacier, 7 = Bagshawe Glacier). The locations for sediment cores collected in January 2016 and included in this study are indicated by the star. The dashed yellow line indicates the transect along which vertical sections are plotted. Blue outline (inset right) shows glacial fronts where meltwater is introduced in the model.**

A total of 18 stations per season were sampled for Fe geochemical variables using acid-cleaned 12 L GO-Flo bottles (General Oceanics) suspended in series on a clean hydroline (Amsteel) and triggered with acid-cleaned Teflon messengers designed by Ken Bruland (UC Santa Cruz). This sampling effort coincided with concurrent CTD stations. Once on board, GO-Flo bottle tops and bottoms were covered with plastic and placed on a wooden rack located within the trace metal clean shipboard plastic "bubble", which was positively-pressurized with HEPA-filtered air. Samples for dFe analysis were pressure-filtered (high purity N2 gas) directly from GO-Flo bottles through 0.2 μm Acropak 200 capsule filters (VWR International), into low-density polyethylene bottles (Nalgene) and acidified to pH 1.7 to 1.8 using HCl (Optima grade, Fisher Scientific). Samples for Fe-binding ligands were similarly filtered in-line but collected in fluorinated high-density polyethylene (Nalgene) bottles, unacidified, and frozen at -20ºC until laboratory analysis back on land. In brief, sample bottles were soaked in a soap detergent overnight with heat applied (60ºC), followed by a one-week soak in 3N HNO₃ (trace metal grade) at room temperature, and finally, a one-week soak in a 3N HCl (trace metal grade) bath at room temperature. Rinsing with ultrapure MilliQ water occurred after each step. This sampling protocol followed established trace-metal clean methods to the standards of the GEOTRACES program to avoid metal contamination. In addition to the filtered samples, unfiltered seawater was sampled directly from the GO-Flo bottles and acidified to pH 1.8 and stored for >6 months (up to 2 years) and filtered prior to analysis in order to determine total dissolvable Fe (TDFe). Labile particulate Fe (LpFe) is





calculated as the difference between TDFe and dFe. Prior to analysis in the laboratory, these unfiltered acidified samples were vacuum filtered using acid-cleaned 0.4 μm polycarbonate (PC) filters in a Teflon filtration apparatus. Particulate samples were collected on 0.4 μm PC filters and stored at -20°C until complete digestion using an $HNO_3/HF$ mixture. The digestion method employed is described in Planquette and Sherrell (2013) and was applied to the US GEOTRACES GP16

total particulate trace metal sample set (Fitzsimmons et al., 2017).

Acute attention to cleanliness was applied when sampling icebergs during small boat deployments in the fjord. Floating icebergs were sampled using a clean stainless-steel pickaxe and rust-free stainless-steel screwdriver and plastic mallet for chiseling pieces of ice. Samples were collected by slowly (engine idled) approaching the target piece of floating ice from downwind, limiting the chance of engine exhaust contamination. Each piece of ice was collected above freeboard (sea

surface), to reduce the chance the ice was altered by seawater and rinsed with MilliQ prior to placing into acid-cleaned 2 gallon Ziploc polyethylene bags and storing at -4°C until sample processing. Prior to filtration, ice samples were removed from the freezer and left to melt at ambient shipboard temperatures. Once completely melted, a small incision was made on the Ziploc bags using a clean stainless-steel razor and contents poured into the Teflon filtration manifold or directly into sample bottles, thus collecting samples for dissolved, total dissolvable and particulate trace metal fractions.

**2.2 Trace metal concentrations**

Stored acidified filtered seawater samples were analyzed for Fe at Scripps Institution of Oceanography using flow injection with chemiluminescence methods described by Lohan *et al.* (Lohan et al., 2006). Dissolved Fe in the samples was oxidized to iron(III) for 1 h with 10 mM $Q-H_2O_2$, buffered in-line with ammonium acetate to pH ~3.5 and selectively pre-concentrated on a chelating column packed with a resin (Toyopearl® AF-Chelate-650M). Dissolved Fe was eluted from the column using

0.14 M HCl (Optima grade, Fisher Scientific) and the chemiluminescence was recorded by a photomultiplier tube (PMT, Hamamatsu Photonics). The manifold was modified based on Lohan *et al.* (2006). Standardization of Fe was carried out with a matrix-matched standard curve (0, 0.4, 0.8, 3.2, 10 nmol kg$^{-1}$ added high purity Fe metal ICP spectrometry standard in 2% $HNO_3$) using low-Fe open ocean seawater. Standards were treated identically to samples. Accuracy was assessed by repeated measurements of GEOTRACES coastal and Pacific Ocean reference seawater samples. Our measurements of GSC gave Fe

= 1.391±0.115 (*n* = 19, over a three-month period, consensus 1.535±0.115). Our measurements of GSP gave Fe = 0.164±0.024 (*n* = 8, over a one-month period, consensus 0.155±0.045). Consensus values are from the most recent July 2019 compilation (geotraces.org). Precision, determined by replicated analyses of an in-house large-volume reference seawater sample within each analytical session, was typically ±5% or better. For the duration of these analyses, the average LOD (defined as 3x the standard deviation of the blank) was 0.036 (*n* = 10).

A subset of the seawater samples and all freshwater samples were run for Fe and Mn at Rutgers University using isotope dilution-inductively coupled plasma mass spectrometry (ICP-MS) methods based on Lagerström *et al.* (2013) and similar to those described in Annett *et al.* (2017). Briefly, 10 mL aliquots of seawater samples were extracted using a commercially





available automated SeaFAST pico system (Elemental Scientific, Inc.) after online buffering to pH approximately 6.5 using ammonium acetate buffer, achieving a 25-fold pre-concentration after column elution in 0.4 mL 1.6 M ultrapure nitric acid

(Optima grade, Fisher Scientific)(Lagerström et al., 2013). Isotope dilution was used to standardize Fe, while Mn was standardized using external matrix-matched standard treated identically to samples. The analysis of the concentrate was performed on an Element 2 sector-field ICP-MS (Thermo Fisher Scientific). Accuracy and precision (±3%, 1SD, for Fe and Mn) was assessed by repeated measurements of in-house large-volume reference seawater samples within each analytical session. Blanks averaged 51 pmol kg$^{-1}$ for Fe ($n$ = 59; LOD = 48 pmol kg$^{-1}$) and 4 pmol kg$^{-1}$ for Mn ($n$ = 69; LOD = 4 pmol

kg$^{-1}$) for all analytical runs. A comparison of the seawater analysis methods employed here is shown in Fig. S1. In general, there is good agreement (average 11% and 6% difference late Spring and Fall, respectively) between the chemiluminescence and ICP-MS methods, comparable to the uncertainty of GEOTRACES consensus values from the intercalibration of 13 trace metal laboratories (for Fe, RSD 10%, https://www.geotraces.org/standards-and-reference-materials/). Total dissolvable trace metals and particle digests were analyzed using direct-injection ICP-MS methods using external standards and added In as a

matrix and instrument drift corrector for the quantification of particulate Fe, Mn, aluminum (Al), and titanium (Ti) concentrations (Annett et al., 2017).

### 2.3 Sediment cores and diffusive flux

Cores for this study were collected using a 12-barrel Megacore multi-coring device aboard the R/V *Nathaniel B. Palmer* cruise NBP16-01 in January 2016. See Taylor *et al.* (2020) for a complete account of coring efforts and Komada *et al.*

(2016) for a description of the pore water sampling procedures. Porewater dFe and dMn was determined colorimetrically using the ferrozine and formaldoxime techniques, respectively (Armstrong et al., 1979; Burdige and Komada, 2020). For dFe, hydroxylamine-HCl (0.2% final concentration) was added to the samples before analysis, to reduce any dissolved Fe(III) in the samples to Fe(II). For dMn, a solution of hydroxylamine solution was added to an acidified (pH ~1-2) sample, and an EDTA solution was added to remove interference from a colored Fe complex. Porewater oxygen concentrations were

measured using a polarographic microelectrode (Brendel and Luther 1995; Luther *et al.* 1998, 2008). A sequential extraction technique (Goldberg et al., 2012; Poulton and Canfield, 2005) was used to determine sediment Fe speciation for the following fractions: Fe$_{ox}$ (highly reactive, poorly crystalline iron oxides), Fe$_{mag}$ (magnetite), Fe$_{prs}$ (Fe in poorly reactive sheet silicates), Fe$_T$ (total sediment Fe), Fe$_{pyr}$ (Fe in pyrite), and finally Fe$_U$ (unreactive pool under all treatments = Fe$_T$ – (Fe$_{ox}$ + Fe$_{mag}$ + Fe$_{prs}$ + Fe$_{pyr}$)). All extracts were analyzed for Fe by flame Atomic Absorption Spectrometry (for details see Burdige

and Komada 2020).

In this study, we investigate the potential for efflux of dissolved trace metals as a source to the overlying water column. Using equation 1, we can estimate the approximate sediment diffusive flux (J$_{sed}$) for dissolved porewater species.

$$J_{sed} = -\phi D_{sed} \frac{dC}{dz} \qquad (1)$$





In this equation, $\phi$ is the porosity of the sediments, and was found to be on average 0.9 near the sediment surface. Porewater
analyses of dissolved Fe and Mn in the Outer Basin (OB) cores reveal high variability in the top-of-core gradient $\left(\frac{dC}{dz}\right)$ in
porewater Fe and Mn (Fig. S2). An average of two cores gives a gradient of ~21.9 µM cm$^{-1}$ dissolved Fe and ~3.6 µM cm$^{-1}$
dissolved Mn. Assuming a diffusion coefficient for Fe and Mn in free solution for seawater ($D_{SW}$) at 0°C to be 3.15x10$^{-10}$ m$^2$
s$^{-1}$ for Fe(II) and 3.02x10$^{-10}$ m$^2$ s$^{-1}$ for Mn(II), we can then estimate the diffusion coefficient in the sediments ($D_{sed}$) by the
following relationship (van Duren and Middelburg, 2001; Halbach et al., 2019):

$D_{sed} = \frac{D_{SW}}{1-2ln\phi}$ (2)

## 2.4 Iron-binding ligands

A subset of seawater samples was analyzed for dFe-binding ligands using single analytical window methods. The methods
applied here are described extensively in Buck *et al.* 2016 (Buck et al., 2018). Briefly, natural seawater samples were titrated
with dFe (0-35 nM) in order to fully saturate the natural ligands. Following a 2 hour equilibration with the added Fe, a well-
characterized ligand (salicylaldoxime, SA) was added to compete with natural dFe-binding ligands. The concentration of SA
used in this study to examine ligands was 25.0 µmol L$^{-1}$ ($\alpha_{Fe(SA)_x}$ = 115). After at least 15 minutes of equilibration, the
Fe(SA)$_x$ electroactive complex was measured using adsorptive cathodic stripping voltammetry (ACSV) on a hanging
mercury drop electrode (BioAnalytical Systems, Incorporated). Peak heights were measured using ECDSOFT and sensitivity
was optimized in ProMCC (Omanović et al., 2015). A combination of traditional linearization techniques was used to
determine the concentrations and strengths of natural ligands within the seawater sample using ProMCC (Omanović,
Garnier, and Pižeta 2015). The uncertainty on modeled complexation parameters was optimized using single or multiple
ligand fitting. These methods were applied successfully to the GEOTRACES speciation data sets (Buck et al., 2015, 2018).
We calculate the capacity for the free ligand pool to bind Fe at equilibrium (Fitzsimmons et al., 2015), or $\alpha_{FeL'}$, defined as:

$\alpha_{FeL'} = 1 + ([eL] \times K),$ (3)

where $eL$ is the difference between the total ligand concentration ($L_t$) and the dFe concentration, and $K$ is the conditional
stability constant.

## 2.5 Numerical model simulations

Based on Hahn-Woernle *et al.* (2020), the ocean in the Andvord Bay region is modeled with the primitive-equation, finite-
difference Regional Ocean Model System (ROMS, Haidvogel *et al.,* 2008). The grid has a horizontal resolution of ~350 m
and a terrain-following vertical coordinate system with 25 depth layers. Due to the changing terrain, the fixed number of
layers, and surface intensified resolution, the maximum thickness for deeper layers is 84.6 m and the minimum thickness for
surface layers is 0.5 m (to better resolve e.g. the surface currents). The domain has 3 open boundaries: the western end of





Bismark Strait, a passage to the continental shelf in the northwest, and along Gerlache Strait to the northeast (Fig. 1).
Boundary and initial conditions were derived from CTD and ADCP observations. The model is forced with tidal and
meteorological data (from TPXO8 Egbert and Erofeeva 2002 [updated] and RACMO van Wessem *et al.,* 2014, respectively)
and run from November 2015 for 5 months. After one month, the transient effects, based on dynamics and thermodynamics,
were found to no longer be present, and the system was consistent. Only the final four sea-ice free months were analyzed
(December through March). Processes like melting of icebergs and floating sea ice are not modeled directly, therefore such
local freshwater sources are captured in a surface intensified meltwater input applied along the glacial boundaries. These
new freshwater sources include also surface runoff and local melt of glacial ice, while precipitation and snowfall are
represented in the meteorological forcing. The applied volume transport of meltwater is a rough estimate based on few
modeled results and observed data and results in an intensified meltwater input at the glacial fronts (for further details see
Hahn-Woernle *et al.,* 2020). To represent the seasonal cycle of temperature-induced melting the volume flux of inflowing
meltwater follows a bell-shaped temporal distribution peaking at the end of January.

We use this model to identify the potential supply pathways and estimate the hydrographic export of three Fe-rich sources in
Andvord Bay: surface glacial meltwater, subsurface subglacial plume, and deep water masses located within the inner basin.
For this purpose, we designed three model experiments with numerical "dyes" to track potential iron pathways: one, to track
the current seasonal input of meltwater from glaciers in Andvord Bay (*surface meltwater dye experiment*) released along the
glacial fronts in the inner fjord at 0-50 m depth (Fig. 1); and two additional experiments involving subsurface water masses
in front of Bagshawe Glacier in Inner Basin A (IBA, 64° 53' 36'' S, 62° 34' 48'' W) at two different depths, one at ~100 m
and the other at ~300 m (*subsurface* and *deep dye experiments*, respectively). Due to the model geometry, the mean depths
the subsurface and deep dyes were released were 107 (94-120 m) and 314 m (290-342 m), respectively. Covering two
horizontal grid cells each (with different thickness), the subsurface and deep dyes had initial volumes of 5 x $10^6$ and 11.3 x
$10^6$ $m^3$, respectively. It follows from the experiment setup that the meltwater dye has a continuous source while the total
amount of the other two dyes is a constant as long as they do not leave through the open boundaries of the model domain.

## 3 Results

We present seasonal results of Fe and Mn concentration and speciation, including a first assessment of Fe-binding ligands in
a cold-based Antarctic fjord. Using porewater measurements on sediment cores collected in the fjord, we also present
porewater Fe speciation and estimate the sedimentary efflux of dFe and dMn. Finally, the dispersal of Fe-rich sources is
modeled to identify pathways for Fe supply and important dynamics contributing to their dispersal.

### 3.1 Seasonality and hydrography in Andvord Bay

In Andvord Bay (Fig. 1), seasonal changes in phytoplankton biomass were documented, as indicated by the proxy
Chlorophyll-a, which shows a 10-fold concentration decrease across all taxonomic classes between the late spring and fall



cruises (Pan et al., 2020). Associated with these changes in primary production, depletion of the surface macronutrients
nitrate (N) and silicic acid (Si) were observed (Ekern, 2017). Increased Si concentrations within the inner fjord could be
driven by sedimentary processes, or weathering of the bedrock by contact with the 11 marine-terminating glaciers feeding
into Andvord Bay (Hawkings et al., 2018; Ng et al., 2020). Surface stocks of macronutrients were never exhausted (Fig. 2).
The phytoplankton community was dominated by small size classes, with very few large diatoms (Pan et al., 2020). The
microplankton class was sparingly present in the Fall, however, benthic cameras captured a large sedimentation event of
marine aggregates indicative of a large diatom bloom in late-January. The export of biogenic particles from the surface also
showed a distinct seasonality indicated by increased Chlorophyll-a pigment content in seafloor sediment cores (Ziegler et al.,
2020), as well as higher respiration rates from chamber incubation experiments in the Fall compared to Spring (data not
shown), although no indication of sulfate reduction was observed in sediment box and Kasten cores (2.3 m long), suggesting
that oxygen, nitrate, and metal oxides were sufficient to oxidize organic matter within the upper sediments (C. Smith pers.
comm.).







**Figure 2. Seasonal phytoplankton, macro-, micronutrient, temperature, and meltwater distributions plotted as sections extending from the inner basin (IB, left) towards Gerlache Strait (GS, right). Plots were made with Ocean Data View visualization software (Schlitzer, 2002, Ocean Data View, last access: 1 February 2021).**




Derived glacial meltwater fractions (MWf, Fig. 2), based on salinity and oxygen isotopes of seawater, ranged from 0.75-2% in late Spring, and from 0.5-2.5% in the Fall (Pan *et al.*, 2019). The fjord also exhibited a gradient in meltwater content, with highest MWf at the glacier terminus. Using a simple mass balance for the surface layer in Andvord Bay, we estimate an approximate meltwater input of 23600 m$^3$ d$^{-1}$ in order to account for the observed changes in oxygen isotope ratios. This

estimate is within the derived estimates of surface meltwater flux generated by warm atmospheric temperatures ($1.4 \times 10^4$ to $1.2 \times 10^5$ m$^3$ d$^{-1}$; Lundesgaard *et al.*, 2020). MWf is strongly correlated with phytoplankton abundance within Andvord Bay; for a detailed discussion see Pan *et al.* (2019). We find that glacial meltwater impacts phytoplankton within the fjord, but the geographical influence of meltwater can extend across the shelf, hundreds of kilometers from the coastal inputs (Dierssen et al., 2002; Meredith et al., 2017).

Physical properties measured in the study region showed the dominant water masses in the fjord were Antarctic Surface Water (cold fresh) and Bransfield Strait water (cold and salty) (Lundesgaard et al., 2020). However, during late Spring, greater influence of modified Upper Circumpolar Deep Water was observed outside of the fjord, indicated by its distinctly higher temperature at depth, but this water mass is prevented from entering the fjord due to a shallow sill near the fjord mouth in the Gerlache Strait (Fig. 2). Optical measurements recorded a change in the particle concentration and assemblage

between the two cruises. Profiles of beam attenuation coefficient and particulate backscattering coefficient showed strong seasonality (see Fig. 4 and discussion in Pan *et al.,* 2019). Pan *et al*. interpreted these optical signatures in the upper water column as a change from a surface biogenic-dominated assemblage in late Spring to a subsurface lithogenic-dominated assemblage in the Fall, composed of fine suspended particles contained within plumes. An important feature observed within the fjord was a subsurface neutrally-buoyant plume (~100 m) characterized by a point source of relatively cold and particle-

laden water emanating from the terminus of Bagshawe Glacier and extending several kilometers over the inner basin (Fig. S3).

Strong buoyant plumes can drive circulation in fjords via the "meltwater pump", but without estimates of volume flux at the glacier grounding line, it is not possible to determine the effect of small amounts of basal and subglacial melt on circulation in Andvord Bay. While this process is described in-depth for Arctic glaciers, Andvord Bay differs in that ocean temperatures

are approximately -1 °C at depth, too cold to ablate the glacier terminus, and neutral buoyancy is reached below the surface layer (indicated by subsurface sediment plumes, Domack and Ishman 1993). However, two important consequences of these plumes are a flux of suspended particulate matter within subsurface "layers" as indicated by high beam attenuation coefficient and optical backscatter (Fig. S3 in Pan *et al.*, 2019), and general mid-water cooling found in the inner fjord (Figure 8 in Lundesgaard *et al.*, 2020). Downstream mixing mechanisms, such as flow over topographic features or wind

induced upwelling, could displace plume water closer to the euphotic zone.



## 3.2 Water column trace metals

Dissolved Fe concentrations in the surface, defined as the upper ~20 m based on similar mixed layer depths (MLD) for both seasons (Lundesgaard *et al.*, 2020), changed seasonally with an overall increase in dFe concentration in the Fall (Fig. 3). The average surface concentration during late Spring was 2.47±0.92 nM ($n$ = 21), while in Fall it was 6.67±1.41 nM ($n$ = 19).

Water column trace metals are presented in Table S1. These concentrations are within the ranges of dFe determined in prior studies (1-31 nM) in the northern WAP region but indicate that large temporal variability exists in surface waters in this region (Hatta *et al.* 2013; Sanudo-Wilhelmy *et al.* 2002; Ardelan *et al.* 2010; Martin *et al.* 1990). The smaller range of surface concentrations during late Spring suggests that dFe was more tightly controlled by phytoplankton uptake, whereas in the Fall, patchiness among stations arises due to varying proximity to Fe sources and the effects of circulation and mixing.

Vertical profiles of dFe showed a steep increase to values greater than 10 nM at the deepest depths sampled during late Spring, especially at stations located within the inner fjord and basins (Fig. 2, 4). In the subsurface (50-150 m), an enriched dFe source was present with average concentrations 3.68±1.52 nM in late Spring and 7.38±2.49 nM in the Fall. Deep water masses greater than 150 m deep had the highest average concentrations of dFe but a seasonal decrease in concentration was observed (8.79±4.75 nM in late Spring, 6.37±2.38 nM in Fall). The greatest concentrations of dFe were found in the inner

fjord and basin stations, with the exception of one station located at the mouth of the fjord near Aguirre Channel (station AC in Fig. 1). Water column concentrations were lower in the Gerlache Strait and fjord mouth. The general shapes of the profiles in late Spring are characteristic of a stratified water column, with dramatic ferriclines below the surface.





**Figure 3.** Surface (<20m) dissolved Fe (top) and meltwater fraction (bottom) for late Spring (left two panels) and Fall (right two panels). Plots were made with Ocean Data View visualization software (Schlitzer, 2002, Ocean Data View, last access: 1 February 2021).





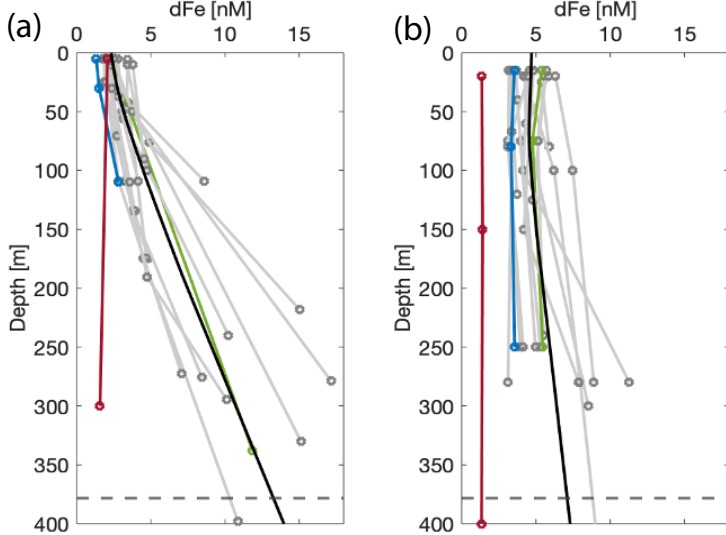

**Figure 4. Depth profiles of dissolved Fe [nM] sampled in the Andvord Bay region for December 2015 (A) and April 2016 (B). The colored lines indicate highlighted profiles: the geometric mean of the linearly interpolated data points within Andvord Bay (black), Station B on the continental shelf (red, see Fig. 1), Station GS (Gerlache Strait, blue) and Station S3 (green). Other Andvord Bay stations are shown in grey. The dashed line is the average bottom depth within the fjord.**

In the Fall, surface dMn was more than double that observed in the late Spring, but surface dFe showed a greater seasonal increase, such that the dissolved Mn:Fe ratio decreased overall and was more variable than in late Spring, when concentrations of dMn remained below 4.5nM, even at depth. Labile particulate Mn (LpMn = TDMn – dMn) showed strong co-variation with LpFe and beam attenuation coefficient c(660). The comparatively high surface dissolved Mn:Fe ratios in late Spring were presumably due to intense biological drawdown of Fe during the vernal bloom, evidenced from low concentrations of dFe where phytoplankton biomass (as Chl-a) was highest (Fig. 5a). In the late Spring, dFe is anti-correlated with MWf (Fig. 5c), whereas there was no significant trend between dFe, biomass and MWf variables in the Fall (Fig. 5b,d). The correlation between dMn and dFe was stronger in the Fall, however, compared to the late Spring (Fig. 5e,f). Labile particulate iron (LpFe = TDFe - dFe) concentrations were elevated in the inner basins in the late Spring and Fall, and strongly correlated with suspended particle concentrations, indicated by optical beam attenuation coefficient c(660) m$^{-1}$ (Fig. 5n, 6). Average LpFe concentrations in the surface were comparable to surface waters in Ryder Bay (southern Antarctic Peninsula), where TDFe varied temporally from 57 to 237 nM (Annett et al., 2015). This comparison between LpFe and TDFe is valid since TDFe is much greater than dFe in these two coastal locations, hence it is a good approximation of LpFe. The LpFe maxima were associated with high turbidity in the inner basins, reaching as high as 900 nM at 300m depth in the Fall (Fig. 6). Dissolved Fe and LpFe were highly correlated (r$^2$ = 0.48 late Spring $n$ = 19; 0.77 Fall $n$ = 28), implying active exchange between these pools (Fig. 5g,h). On average, dFe made up 3.1% (late Spring) and 4.6% (Fall) of the total





dissolvable pool. The LpMn concentrations displayed similar seasonality to LpFe and similar association with total particles,

but were more strongly correlated in the Fall (Fig. 5l). Dissolved Mn and LpMn were highly correlated ($r^2 = 0.70$ late Spring $n = 19$; 0.79 Fall $n = 28$; Fig. 5i,j). On average, dMn composed 52% (late Spring) and 57% (Fall) of the total dissolvable pool.

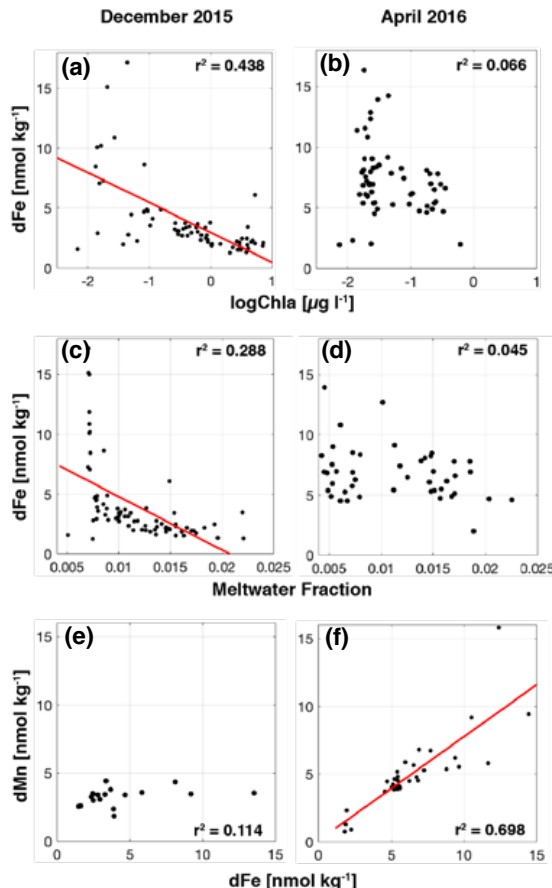

**Figure 5. Dissolved trace metals plotted against observed and derived variables for December 2015 (a, c, e) and April 2016 (b, d, f).**
**Dissolved Fe (a-b) versus logChlorophyll-a concentrations. Dissolved Fe (c-d) versus meltwater fraction. Dissolved Mn (e-f) versus dissolved Fe. Least-squares regression lines are shown where they are statistically significant ($p < 0.005$).**

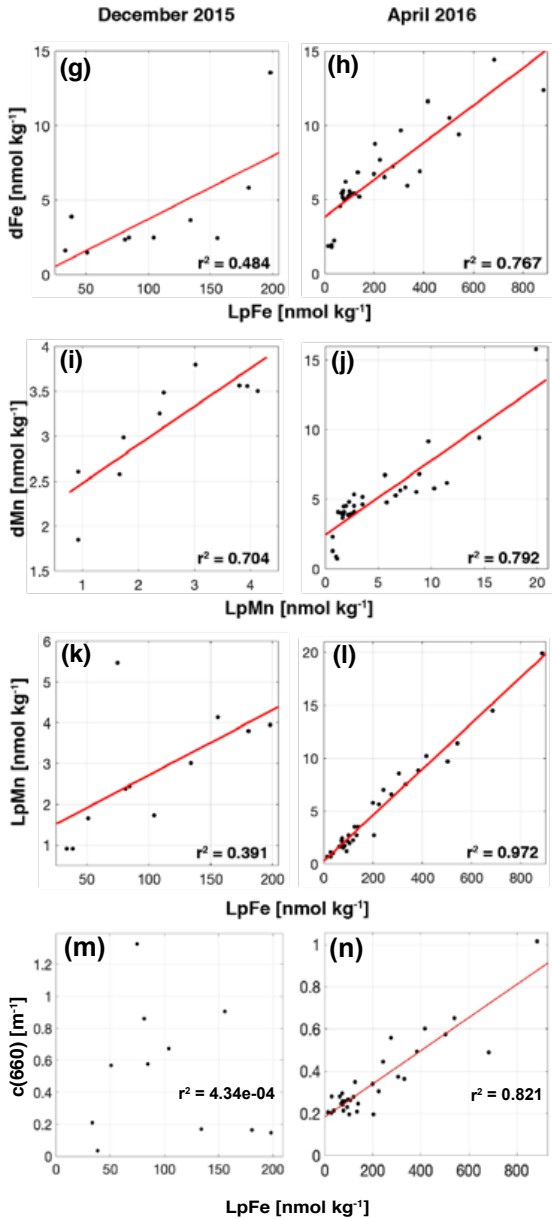

**Figure 5 (cont'd). Dissolved Fe and Mn concentrations versus labile particulate Fe and Mn for each season. Dissolved Fe (g-h), labile particulate Mn (k-l), and beam attenuation coefficient (m-n) versus labile particulate Fe. Dissolved Mn (i-j) versus labile particulate Mn. Least-squares regression lines are shown where they are statistically significant ($p < 0.005$).**



**Figure 6. Total dissolvable trace metals and beam attenuation coefficient c(660) for both seasons. The transects are plotted as distance from the Bagshawe Glacier terminus. Plots were made with Ocean Data View visualization software (Schlitzer, 2002, Ocean Data View, last access: 1 February 2021).**


### 3.3 Glacial ice and plume trace metals

Glacial ice and plume samples were analyzed for Fe, Mn, Al, and Ti concentrations, which are presented in Table 1. Three glacial ice samples were analyzed for dFe ($71.52\pm121.31$ nM) and dMn ($49.43\pm82.64$ nM). Visual inspection of Glacial Ice 3 and 4 showed these pieces contained low particle loads, while Glacial Ice 1 and 2 had a comparatively high content of dark

colored coarse-grained particles. Hence, these and the "clean" glacial ice samples are indicative of the variability of trace metal concentrations in icebergs found in Andvord Bay. Labile particulate trace metal concentrations were two orders of magnitude higher than the dissolved fraction based on two ice samples ($40.71\pm85.58$ µM LpFe, $3.64\pm5.06$ µM LpMn). We did not determine labile particulate trace metals for Glacial Ice 3 and 4, thus these average labile particulate concentrations





are skewed toward a high value. Total particulate trace metals showed similar concentration variability to the dissolved

fraction (94.87±181.08 µM TpFe, 2.66±5.06 µM TpMn). For Glacial Ice 3 and 4, the concentration of dMn was greater than

TpMn. The ratios of labile and total particulate Mn:Fe were 0.061±0.002 mol:mol and 0.028±0.004 mol:mol, respectively.

Dissolved Al and Ti were not analyzed for these ice samples, but total dissolvable and total particulate samples were

analyzed for Glacial Ice 1 and 2, and 1-4, respectively. We defined the refractory particulate trace metal concentration as the

difference between the total particulate and total dissolvable fractions (RpTM = TpTM - TDTM). Total dissolvable Al and

Ti average concentrations were skewed due to the heavy particle load present within Glacial Ice 1 and 2 (603.2±715.68 µM

TDAl, 20.75±27.06 µM TDTi). Total particulate Al and Ti had similar variability to the total dissolvable fraction and

included all four glacial ice samples with averages of 428.11±790.46 µM TpAl and 13.41±25.68 µM TpTi, therefore the

average total particulate concentrations were lower than the average determined for total dissolvable Al and Ti in Glacial Ice

1 and 2. We found the labile particulate concentration to be a valid comparison to total dissolvable since dFe concentration

was on average 1.8±1.5% of TpFe concentration. Thus, the particulate fraction dominated trace metal speciation of total Fe,

Mn, Al, and Ti in glacial ice.



| Sample Type | Cruise | Location | | dFe [nmol kg-1] | TDFe [nmol kg-1] | TpFe [nmol L-1] | dMn [nmol kg-1] | TDMn [nmol kg-1] | TpMn [nmol L-1] | dFe:LpFe | dFe:TpFe | LpMn:LpFe [mol:mol] | TpMn:TpFe [mol:mol] | Description |
|---|---|---|---|---|---|---|---|---|---|---|---|---|---|---|
| | | | | | | | | | | | | (0.017) | (0.017) | Crustal averages |
| Glacial Ice 1 | LMG1510 | Inner Basin A | | 211.59 | 122228.21 | 366332.00 | 144.85 | 7359.13 | 10187.59 | 0.17% | 0.1% | 0.059 | 0.028 | floating |
| Glacial Ice 2 | LMG1510 | Inner Basin A | | n.a. | 994.26 | 13036.31 | n.a. | 61.55 | 434.61 | n.a. | n.a. | 0.062 | 0.033 | floating |
| Glacial Ice 3 | NBP1603 | Neko Harbor | | 1.43 | n.a. | 56.36 | 1.62 | n.a. | 1.41 | n.a. | 2.5% | n.a. | 0.025 | floating |
| Glacial Ice 4 | NBP1603 | Neko Harbor | | 1.52 | n.a. | 52.75 | 1.81 | n.a. | 1.26 | n.a. | 2.9% | n.a. | 0.024 | floating |
| | | | AVG | 71.52 | 61611.24 | 94869.36 | 49.43 | 3710.34 | 2656.22 | 0.17% | 1.8% | 0.061 | 0.028 | |
| | | | STDEV | 121.31 | 69994.45 | 181078.54 | 82.64 | 5160.17 | 5025.07 | | 1.5% | 0.002 | 0.004 | |
| Plume 19_100m | NBP1603 | Inner Basin A | | 11.64 | 419.89 | 402.26 | 5.82 | 15.73 | 9.169 | 2.9% | 2.9% | 0.024 | 0.023 | seawater |
| Plume 20_100m | NBP1603 | Inner Basin A | | 9.40 | 538.34 | 514.90 | 6.22 | 17.29 | 11.73 | 1.8% | 1.8% | 0.021 | 0.023 | seawater |
| Plume 26_110m | NBP1603 | Inner Basin A | | 7.26 | 227.24 | 330.37 | 5.28 | 11.63 | 7.665 | 2.7% | 2.2% | 0.024 | 0.023 | seawater |
| Plume 27_110m | NBP1603 | Inner Basin A | | 6.72 | 202.35 | n.a. | 4.79 | 10.36 | n.a. | 3.4% | n.a. | 0.028 | n.a. | seawater |
| | | | AVG | 8.75 | 346.95 | 415.84 | 5.52 | 13.75 | 9.52 | 2.7% | 2.3% | 0.024 | 0.023 | |
| | | | STDEV | 2.25 | 160.40 | 93.01 | 0.62 | 3.29 | 2.05 | 0.7% | 0.5% | 0.003 | 0.0002 | |

| Sample Type | Cruise | Location | | TDAl [nmol kg-1] | TpAl [nmol kg-1] | TDTi [nmol kg-1] | TpTi [nmol kg-1] | TpFe:TpAl [mol:mol] | RpFe:RpAl [mol:mol] | TpFe:TpTi [mol:mol] | LpFe:TDAl [mol:mol] | TDAl:TDTi [mol:mol] | TpAl:TpTi [mol:mol] | Description |
|---|---|---|---|---|---|---|---|---|---|---|---|---|---|---|
| | | | | | | | | (0.2) | (0.2) | (7) | (0.2) | (35) | (35) | Crustal averages |
| Glacial Ice 1 | LMG1510 | Inner Basin A | | 1109261.83 | 1611691.07 | 39878.61 | 51906.02 | 0.23 | 0.49 | 7.06 | 0.11 | 28 | 31 | floating |
| Glacial Ice 2 | LMG1510 | Inner Basin A | | 97132.07 | 100238.65 | 1614.41 | 1708.99 | 0.13 | 3.88 | 7.63 | 0.01 | 60 | 59 | floating |
| Glacial Ice 3 | NBP1603 | Neko Harbor | | n.a. | 259.19 | n.a. | 6.56 | 0.22 | n.a. | 8.59 | n.a. | n.a. | 40 | floating |
| Glacial Ice 4 | NBP1603 | Neko Harbor | | n.a. | 262.50 | n.a. | 6.58 | 0.20 | n.a. | 8.02 | n.a. | n.a. | 40 | floating |
| | | | AVG | 603196.95 | 428112.85 | 20746.51 | 13407.04 | 0.19 | 2.18 | 7.82 | 0.06 | 44 | 42 | |
| | | | STDEV | 715683.82 | 790458.42 | 27056.88 | 25678.53 | 0.04 | 2.40 | 0.40 | 0.07 | 23 | 12 | |
| Plume 19_100m | NBP1603 | Inner Basin A | | 887.79 | 1719.11 | 14.57 | 44.45 | 0.23 | -0.01 | 9.05 | 0.46 | 61 | 39 | seawater |
| Plume 20_100m | NBP1603 | Inner Basin A | | 965.50 | 2110.48 | 13.74 | 56.07 | 0.24 | -0.01 | 9.18 | 0.55 | 70 | 38 | seawater |
| Plume 26_110m | NBP1603 | Inner Basin A | | 828.98 | 1373.37 | 13.85 | 34.72 | 0.24 | 0.11 | 9.52 | 0.33 | 60 | 40 | seawater |
| Plume 27_110m | NBP1603 | Inner Basin A | | n.a. | n.a. | n.a. | n.a. | n.a. | n.a. | n.a. | n.a. | n.a. | n.a. | seawater |
| | | | AVG | 894.09 | 1734.32 | 14.06 | 45.08 | 0.24 | 0.03 | 9.25 | 0.44 | 64 | 39 | |
| | | | STDEV | 68.48 | 368.79 | 0.45 | 10.69 | 0.01 | 0.07 | 0.24 | 0.11 | 6 | 1 | |

**Table 1. Glacial ice and seawater samples analyzed for dissolved, labile, and total particulate trace metals. Crustal averages from Taylor and McClellen (1995): Mn:Fe (0.017 mol:mol), Fe:Al (0.2), and Al:Ti (35).**

Four seawater samples were collected from 100-110 m depth, corresponding to the core of the subsurface turbidity plume within IBA. Average concentrations of dissolved metals were 8.75±2.25 nM dFe and 5.52±0.62 nM dMn. LpFe (350.70±147.92 nM) and LpMn (8.23±2.68 nM) were statistically indistinguishable from the total particulate fractions (415.84±93.01 nM TpFe, 9.52±2.05 nM TpMn) within measurement error, including filter splitting and sample distribution uncertainties. The average ratio of labile particulate Mn:Fe was 0.024±0.003 mol:mol. Particles collected from the plume had high concentrations of Al and Ti, but with distinctly different lability from that of Mn and Fe. The TDAl was 894.09±68.48 nM while TpAl was 1734.32±368.79 nM. Similarly, TDTi was 14.06±0.45 nM and TpTi was 45.08±10.69 nM. The total dissolvable Al:Ti ratio was 64±6 mol mol[-1] and the total particulate Al:Ti ratio was 39±1 mol mol[-1]. The Al:Ti ratio is elevated above the crustal ratio (35 mol mol[-1]) in the total dissolvable fraction, suggesting a larger adsorbed fraction for Al than for Ti.



### 3.4 Glacial sediments

Solid phase Fe speciation of one sediment core from the outer basin station (OB, 64° 46' 46'' S, 62° 43' 57'' W, ~500 m, collected in January 2016), showed an enrichment of authigenic Fe oxides at the surface. Chemical treatments of the
sediments with HCl dissolves poorly crystalline Fe oxy(hydr)oxides (ferrihydrite and lepidocrocite), which are found to be 10% of the total particulate Fe of the surface sediments in this location, compared to an average of 2% below 1.5 cm (Fig. S4). In the surficial sediments, a larger portion of the Fe is associated with poorly labile sheet silicates (e.g. structural Fe(III) in clays, 36%), and a comparable fraction is refractory and is not liberated by any of the solution treatments (31%). Other fractions of particulate Fe are associated with more crystalline and thus less labile Fe oxides (goethite, hematite) and the
minerals magnetite and pyrite. Porewater analyses were performed on two OB cores using colorimetric methods, revealing high concentrations of dFe and dMn. Below the well-oxygenated layer (upper ~0.5 cm), but within the upper 10 cm, dFe reaches its peak concentration of 80 µM, while maximum dMn is 6 µM. Down-core from the peak, concentrations tend to decrease for both trace metals, but there is considerable variability between 15 and 25 cm, including several deeper local maxima. The average porewater concentration of dFe in the top 2.5 cm is 26 µM (Fig. S2). There is considerable difference
in the porewater concentrations of the two OB cores indicating bioturbation of the sediments resulting in large variability on small scales. Points excluded from the oxygen profiles were below the detection limit, while several samples were lost from the porewater profiles, represented as gaps in the vertical traces of dFe and dMn.

### 3.5 Fe-binding organic ligands

To gain insight into the speciation of dFe with the fjord, we analyzed seawater samples for Fe-binding ligands and to identify
comparative strengths of organic Fe complexes (See *Methods*). Analysis of the ligands within Andvord Bay shows a down-fjord gradient in both quantity and quality (all ligand data presented in Table 2). In the late Spring, strong ligands ($LogK_{FeL,Fe'}^{cond} \geq 12.0$) were detected in the surface at stations located within the fjord at concentration levels ranging from 4.06±1.74 nM at Inner Basin A (IBA) to 7.27±1.97 nM at Sill 3 (S3), while only weak ligands ($LogK_{FeL,Fe'}^{cond} < 12.0$) were detected in the Gerlache Strait (GS; 5.72±2.21 nM). An excess of strong ligands, relative to dFe, was detected in the inner
basins. A gradient in concentration of undersaturated ligands (eL in Table 2) is observed towards the GS, with increasing eL. Within the fjord, weak ligands were detected at Inner Basin B (IBB), closest to Moser Glacier. In the Fall, total ligand concentrations ($L_t$) were elevated everywhere within the fjord, but the surface ligands were somewhat weaker compared to the late Spring. The greatest concentrations of ligands were found closest to the glaciers (range 11.18 – 15.42 nM) and in the GS (12.00±2.94 nM). For both seasons, weak ligands were detected in the subsurface, but a greater concentration in the Fall
suggested that these ligands have a local source within the fjord. Compared to other stations in the Fall, we found the plume to contain a small excess of weak ligands (IBA, 110 m). Interestingly, the highest concentration of strong ligands (17.44±1.12 nM) among all sites was in deep water of Station IBA, at 280 m. This is the deepest depth sampled for Fe-



binding ligands and the IBA bottom depth was 382 m. We found a down-fjord gradient in ligand strength at the surface, decreasing with distance from the inner basins ($LogK_{FeL,Fe'}^{cond}$ = 11.95 at IBA, 11.03 at GS).

| | Station | Depth [m] | dFe [nM] | Fe' [pM] | $L_t$ [nM] | +/- | logK | +/- | eL [nM] | $L_t$:dFe | log$\alpha_{FeL'}$ | $R_{Fe'}$ |
|---|---|---|---|---|---|---|---|---|---|---|---|---|
| | IBA | 6 | 1.85 | 0.62 | 4.06 | 1.74 | 12.13 | 0.69 | 2.21 | 2.2 | 12.5 | |
| | IBA | 160 | 5.84 | 5.66 | 8.18 | 0.57 | 11.63 | 0.2 | 2.34 | 1.4 | 12 | |
| | IBB | 6 | 3.36 | 1.19 | 6.22 | 0.52 | 11.99 | 0.14 | 2.86 | 1.9 | 12.4 | |
| December 2015 (LMG1510) | MBA | 8 | 2.12 | 4.7 | 2.82 | 0.37 | 11.58 | 0.2 | 0.69 | 1.3 | 11.4 | |
| | S3 | 11 | 3.41 | 0.79 | 7.27 | 1.97 | 12.05 | 0.44 | 3.85 | 2.1 | 12.6 | |
| | S4 | 7 | 2.01 | 2.39 | 5 | 1.22 | 11.44 | 0.33 | 2.99 | 2.5 | 11.9 | |
| | S4 | 175 | 4.76 | 5.95 | 5.05 | 1.06 | 10.9 | 0.28 | 0.29 | 1.1 | 10.4 | |
| | GS | 6 | 1.53 | 6.2 | 2.26 | 0.33 | 11 | 0.14 | 0.73 | 1.5 | 10.9 | |
| | OBB | 6 | 2.5 | 1.16 | 5.72 | 2.21 | 11.82 | 0.44 | 3.22 | 2.3 | 12.3 | |
| | IBA | 25 | 7.8 | 1.15 | 15.42 | 2.82 | 11.95 | 0.26 | 7.62 | 2 | 12.8 | 85% |
| | IBA | 110 | 6.72 | 7.18 | 8.54 | 0.88 | 11.69 | 0.35 | 1.82 | 1.3 | 12 | |
| | IBA | 280 | 14.45 | 1.82 | 17.44 | 1.12 | 12.42 | 0.37 | 2.99 | 1.2 | 12.8 | |
| | IBA | 80 | 8.51 | 3.86 | 17.62 | 3.57 | 11.38 | 0.27 | 9.11 | 2.1 | 12.3 | |
| | IBB | 20 | 6.89 | 4.16 | 11.18 | 1.39 | 11.58 | 0.26 | 4.29 | 1.6 | 12.2 | 249% |
| April 2016 (NBP1603) | IBB | 75 | 5.94 | 3.64 | 14.03 | 1.57 | 11.3 | 0.13 | 8.09 | 2.4 | 12.2 | |
| | MBA | 20 | 4.25 | 18.7 | 13.32 | 4.08 | 10.37 | 0.25 | 9.07 | 3.1 | 11.3 | 297% |
| | S3 | 15 | 5.41 | 15.67 | 15.33 | 2.18 | 11.54 | 0.14 | 9.92 | 3 | 12.5 | 1883% |
| | S4 | 25 | 6.93 | 4.63 | 15.4 | 3.11 | 11.2 | 0.24 | 8.47 | 2.2 | 12.1 | 94% |
| | Fjord Mouth | 15 | 4.69 | 10.05 | 6.29 | 1.29 | 11.43 | 0.55 | 1.6 | 1.3 | 11.6 | |
| | Fjord Mouth | 120 | 5.37 | 6.56 | 7.34 | 1.87 | 11.6 | 0.68 | 1.97 | 1.4 | 11.9 | |
| | GS | 15 | 5.14 | 6.82 | 12 | 2.94 | 11.03 | 0.32 | 6.86 | 2.3 | 11.9 | 10% |


**Table 2. Ligand concentrations and equilibrium constants detected in seawater samples. Fe' is the free (unbound) iron concentration. $L_t$ is the total ligand concentration. logK is the conditional stability constant. eL is the excess ligand concentration (eL = $L_t$ − [dFe]). log$\alpha_{FeL'}$ is the complexation capacity. $R_{FeL'}$ is the ratio of Fe' of reoccupied stations, expressed as a percentage.**

We determined the free (uncomplexed) Fe concentration (Fe' in Table 2) within samples analyzed for Fe-binding ligands. In the surface, a greater concentration of Fe' was found in the Fall (8.74±6.43 pM, *n* = 7) compared to the late Spring (2.44±2.18 pM, *n* = 7). Water below the surface showed similar concentrations for each season (5.8±0.21 pM late Spring, 4.61±2.22 pM Fall). The greatest concentrations of Fe' were observed mid-fjord at the surface (18.7 pM Fe' at MB, 15.67 pM Fe' at S3) in the Fall.



### 3.6 Dye experiments

To study the transport pathways for dFe, we use numerical passive dyes in the Hahn-Woernle *et al.* (2020) regional model of Andvord Bay (see Fig. 1 in Hahn-Woernle *et al.,* 2020) to track three potential sources of dFe: surface glacial meltwater (0-50 m) from Bagshawe and Moser Glacier termini, neutrally-buoyant subsurface plume (100 m), and deep water located in IBA (300 m; as in *Section 2.5*). Due to numerous inputs and complex biogeochemical processes which result in observed dFe distributions in time and space, we simplify the problem by assuming no removal over the duration of simulated dye experiments. We use this approach to illustrate the multiple transport pathways for dFe supply to the fjord and surrounding ocean from December through March (St-Laurent et al., 2017). The results are presented first for the surface meltwater experiment, followed by two fixed-volume experiments, referred to as subsurface and deep dye experiments.

Most of the surface glacial meltwater dye remains in the upper 100 m throughout the model run, and due to its proximity to the surface, it is quickly dispersed over a large region by relatively rapid surface currents. It takes about 10-15 days for the surface meltwater to exit the fjord mouth, where most ends up in the central and northern Gerlache Strait after 120 days (Fig. S5a).

The subsurface dye (100 m) is spread more rapidly than the deep dye (300 m). After 8 days, the subsurface dye reaches the fjord mouth, which is 4 days before the deep dye, implying it has a shorter residence time within the fjord compared to the deep dye. We loosely define residence time as the model timestamp at which a fixed fraction of dye remains within the fjord domain. After 22 days, 25% of the subsurface dye has left the fjord, while it takes the deep dye almost twice as long (43 days). At the end of the 120 days long model run, less than 18% of the subsurface dye and over 30% of the deep dye remain in the fjord domain (Fig. S6a). Looking at the whole model domain in Fig. 1, which includes Andvord Bay and Gerlache Strait, only 59% of the subsurface dye and 75% of the deep dye are still present after 120 days. The missing 41% (25%) has mainly left the model domain through the Gerlache Strait to the north, where these waters mix with Bransfield Strait water and subsequently with the southern Antarctic Circumpolar Front waters.

We analyzed the vertical distribution of the subsurface and deep dyes along the fjord mouth and horizontally over different depth layers. Within the first day, the subsurface dye spreads over the depth range of 20 to 125 m and the deep dye over 125 to 500 m (>1% of dye per depth layer). The subsurface dye leaves the fjord mainly within the upper 200 m. After 8 days, as the subsurface dye reaches the fjord mouth (Fig. S5b), the maximum concentration is still found close to its release depth at 100-125m. Over the next few days, surface layer concentrations (<20m) increase, but the highest concentration is soon found below 125m (after 2 weeks) (Fig. S6a).

The deep dye remains mainly below 200 m as it passes the fjord mouth (maximum water depth at the fjord mouth is 360 m). After 12 days, as the deep dye reaches the fjord mouth, the maximum concentration is found below 300 m depth. In contrast to the subsurface dye, the deep dye remains longer in the proximity of the fjord mouth and on several occasions, re-enters the fjord leading to a longer residence time within the fjord (Fig. S5c). The majority of the deep dye leaves the fjord at depths below 100 m and along the southwestern coastline. Both dyes, subsurface and deep, have low concentrations in the upper



100 m of the northeastern flank of the fjord mouth. This is due to the inflow of external water from the GS along the northeastern coastline. Throughout the run, the deep dye is confined to the inner basins of the fjord. In all cases, the dyes
remain at higher concentrations and for longer periods in the subsurface fjord waters than in the surface layer, which shows faster transport out of the fjord.

## 4 Discussion

### 4.1 Iron sources in a heavily glaciated fjord

Due to the proximity to glaciers and influence of ice within Andvord, we hypothesized meltwaters to be an important source
of Fe. We focus on quantifying dissolved, total dissolvable and particulate Fe and Mn, as well as total dissolvable and particulate Al and Ti. Ratios of these elements are treated as proxies for contributions of various endmembers. Candidate endmembers include reducing sediments, weathered crustal material, and biogenic particles (Taylor and McLennan 1995; Twining *et al.*, 2004). Where possible, we estimate fluxes of dFe. We begin by examining the relationship between glacial meltwater and dFe.

### 4.2 Role of surface glacial meltwater

Glacial meltwater at the surface has the potential to be a significant source of Fe to phytoplankton. There exists a weakly negative correlation between derived MWf and dFe at the start of the melt season (late Spring: $r^2 = 0.29$, n = 30; early-Fall: $r^2 = 0.05$, n = 13; Fig. 5c,d). One possible explanation is that increased meltwater at the surface leads to greater stratification and limits upwelling of Fe-rich deep water, with the effect augmented by removal processes, such as biological drawdown
and scavenging of dFe onto sinking particles. Indeed, higher rates of primary production are associated with greater fractions of meltwater in Andvord Bay (Pan et al., 2020). Since glacial meltwater is restricted to the surface, it constitutes a significant input of Fe to the surface throughout the growth season. While we observe high concentrations of dissolved and particulate trace metals within glacial ice, we note that the icebergs within Andvord were predominantly "clean" ice, with little sediment embedded in the ice, indicated by relatively low dFe and TpFe (for instance, Glacial Ice 3 and 4 in Table 1). Based on Fe:Al
ratios in particles and average values for continental crust (Taylor and McLennan 1995), we estimate $87\pm22\%$ (*n* = 4) of the particulate Fe contained within Andvord icebergs is terrigenous in origin. This is consistent with mechanical weathering of continental crust followed by inclusion of the particles into the ice (freeze-in, Raiswell *et al.*, 2018). Low Fe:Ti and Al:Ti ratios also reflect a continental crust source, but it is worth noting that Glacial Ice 2 had significantly more Mn and Al, relative to continental Fe and Ti. Further, Mn and Al solid speciation suggests there are high concentrations of Mn- and Al-
oxides, which may be formed when crustal material is altered (Raiswell *et al.*, 2018). It is also possible that fjord sediments were the source of particulate matter within Glacial Ice 2, which would correspondingly have higher Mn content (and higher Mn:Fe) than what is found in basal ice interacting with the subglacial environment (Hawkings et al., 2020). Continental crust





material delivered to the ocean would contain a relatively low Mn content compared to Fe (Fe is 4% w/w in crustal material, while Mn is 0.08% w/w, Rudnick and Gao 2013).

Visual inspection suggests that the majority of the ice within Andvord has relatively low concentrations of particles, whereas basal ice, with dark layers of sediment (Glacial Ice 1 in Table 1), will likely skew the average towards high values (Hopwood *et al.*, 2019). A compilation of TDFe in icebergs in Antarctica estimated an average concentration of 24 μM (Hopwood *et al.*, 2019). Our two measurements of LpFe in glacial ice are different (average for this study is 61±70 μM LpFe, *n* = 2) but are within the range of concentrations determined in the previous study. Thus, we use our average

concentration (Table 1) as indicative of the glacial ice composition in Andvord to compute the following meltwater fluxes. Using a range of estimated surface glacial meltwater volume inputs ($2.4 \times 10^4$ m$^3$ d$^{-1}$ for this study based on oxygen stable-isotope mass balance; $1.8 \times 10^4$ to $1.2 \times 10^5$ m$^3$ d$^{-1}$ Lundesgaard *et al.*, 2020; $1.1 \times 10^6$ m$^3$ d$^{-1}$ Hahn-Woernle *et al.*, 2020 including other freshwater sources that are not precipitation) and assuming the input of meltwater is distributed evenly over the fjord surface layer, we calculate fluxes on the order of 15.1 to 704 nmol m$^{-2}$ d$^{-1}$ for dFe and 10.4 to 487 nmol m$^{-2}$ d$^{-1}$ for

dMn. Based on modeling work in this paper, it will become evident that meltwater released to Andvord does not stay within the fjord. Additionally, significant metal loss might result from scavenging processes, transferring Fe to depth on sinking particle surfaces, rendering it inaccessible for phytoplankton uptake. Still, the availability of excess macronutrients within Andvord Bay (Fig. 2) means that substantial increases in the supply of trace metals from glacial meltwater would stimulate growth in the euphotic zone, if light were not limiting.

**4.3 The nature of Fe in subglacial plumes**

The inner basins consistently show higher beam attenuation and particle backscattering coefficients than mid-fjord and shelf stations (see Figure 3 in Supplementary Information in Pan *et al.* 2019). These signals are attributed to ultra-fine suspended sediments (<0.7-0.8 μm). The high particle backscattering coefficient in the surface at all stations in late Spring is due to the high concentrations of biogenic particles associated with the vernal bloom. Inner basins also show local maxima in beam

attenuation coefficients at 70-150 m, as well as approaching the benthic boundary layer (Fig. 6). Sediments that originate near the glacier terminus are carried upward in buoyant turbulent plumes, and spread laterally. This is consistent with the presence of glacial meltwater plumes, or "cold tongues", which originate at the glacier grounding line (described in Domack and Williams 2011), entrain deep water masses, and suspend sediments (Straneo and Cenedese, 2015). Since ocean temperatures remained below 0°C in Andvord (see Fig. 2), there is little to suggest basal melting of the ice, as is observed

further south along the WAP. It appears reasonable on the basis of the evidence given above, that the subsurface plume signature is subglacial in origin.

Total digestion and subsequent analyses of marine particles collected within the plume reveal high concentrations of weathered crustal sediments (82-86% of TpFe, 61-64% of TpMn), and also ingrowth of authigenic particles most likely consisting of precipitated Fe- and Mn-oxide phases (16-18% TpFe, 36-39% TpMn). These results suggest that the origin of

plume particles is a chemically-altered crustal source. Labile particulate Fe is 82-100% of TpFe (Table 1). The Fe:Al and



Fe:Ti in plume particles ($0.24\pm0.01$ mol mol$^{-1}$ and $9.25\pm0.24$ mol mol$^{-1}$, respectively) were elevated above the average crustal ratios (0.2 mol mol$^{-1}$ Fe:Al, 7 mol mol$^{-1}$ Fe:Ti), which implies these samples are enriched in Fe relative to both crustal Al and Ti. In agreement with these results, particulate Al:Ti ($39\pm1$ mol mol$^{-1}$) was elevated above crustal ratios (35 mol mol$^{-1}$), indicating a large oxide fraction is associated with this particulate matter. This substantiates our claim that most of the Fe found in the plume is weakly adsorbed to particles and recently precipitated, since dilute HCl leaches liberate the most labile forms of Fe, most likely as oxy(hydr)oxides (e.g. ferrihydrite) in addition to some Fe from clays. This could include oxides directly precipitated from the anoxic subglacial source, as well as a potential fraction of oxides derived from fjord sediments and porewaters entrained at the grounding line.

Cold-based glaciers are locations where the subglacial environment flows directly into the fjord with minimal mixing with seawater. We find elevated concentrations of dMn emanating from the inner fjord, indicative of the reducing conditions beneath Moser and Bagshawe glaciers, consistent with other studies of subglacial environments (Henkel *et al.*, 2018; Zhang *et al.*, 2015). We report relatively low concentrations of dFe within the plume ($8.75\pm2.25$ nM) <1 km away from the glacier terminus. If we assume a MWf of 0.01 for the plume, the subglacial meltwater endmember would have a dFe concentration of 875 nM, which is higher than the mean value for TDFe measured within the plume ($346.95\pm160.40$ nM) suggesting settling loss through flocculation is likely occurring even within 1 km of the grounding line. The subglacial endmember dFe is lower than the range used to parameterize subglacial inputs from ice shelves to the SO (3 – 30 µM in Death *et al.*, 2014). This perhaps indicates a major difference between glaciers containing large volumes of subglacial meltwater that accumulate the products of more extensive reductive chemical weathering, and smaller glaciers situated on steep topography and which feed into fjords, such as those along the WAP. The long residence time and enhanced chemical weathering beneath large glaciers in west Antarctica (PIG, Thwaites Glacier) could result in large accumulations of dissolved trace metals in subglacial outflow. However, subglacial discharge occurs at some distance from the open continental shelf waters because of the broad floating horizontal ice shelves, which make up about 45% of the Antarctic coastline and can extend 10s – 100s km from the shelf (Schodlok et al., 2016). Our results suggest that assuming such high export efficiency to the coastal ocean (i.e., using endmember concentrations from glacial runoff and groundwaters as in Death *et al.*, 2014) potentially overestimates dFe supply from anoxic subglacial environments because dFe rapidly precipitates after mixing with seawater.

### 4.4 Role of sediments

Analyses of Andvord Bay sediments reveal they are compositionally distinct from temperate fjords consisting of poorly sorted fine silt and clay, many dropstones, suspension deposits and ice-rafted debris (Eidam et al., 2019). Sediment accumulation rates are spatially variable, but a weak along-fjord gradient is present. These deposits suggest sluggish circulation, allowing for the deposition of sediments close to their source, likely through flocculation processes (Cowan and Powell, 1990).

Profiles of beam attenuation coefficient show highest concentration of particles in the inner basins compared to other station locations (see Figure 4 in Pan *et al.,* 2019). There is little evidence for mechanical resuspension through gravity flows (i.e.,





turbidites) along the steep basin walls, yet such processes could be responsible for the near-bottom elevation in water column

particles (Eidam et al., 2019). The presence of elevated particles in the inner basins is accompanied by the greatest concentrations of dissolved and labile particulate Fe and Mn (Fig. 6), demonstrating the potential of resuspended fjord sediments as a source of dissolved trace metals.

Based on the core top porewater profiles, we estimate the sedimentary efflux to be 43.7 $\mu$mol m$^{-2}$ d$^{-1}$ for dFe and 7.2 $\mu$mol m$^{-2}$ d$^{-1}$ for dMn, due to diffusion alone (Fig. S2). This magnitude of flux was also observed in the shelf sediments in the

vicinity of South Georgia Island in the SO (Schlosser et al., 2018). Abundant epibenthic fauna were observed within Andvord Bay, which mix the sediments through bioturbation while consuming labile organic matter. The result is deviation from results based on diffusion alone. Taylor *et al.* (2020) used $^{234}$Th as a proxy to investigate the effect of bioturbation on short timescales and found Andvord Bay sediments possess a high mixing coefficient down to 5 cm ($D_b$ = 36 cm$^2$ yr$^{-1}$) consistent with greater deposition and subsequent utilization of organic carbon in the sediments. We believe this accurately

reflects the conditions in this fjord: bioturbation by dense aggregations of epibenthic fauna within the basins.

These results are not surprising when compared to a global compilation of in situ measurements of sedimentary efflux of dFe, which is on average ~12 $\mu$mol m$^{-2}$ d$^{-1}$ for water masses located on continental margins and with O$_2$ concentrations greater than 63 $\mu$mol L$^{-1}$ (Dale et al., 2015). The bottom water oxygen concentration in Andvord Bay always exceeded 230 $\mu$mol L$^{-1}$. The bottom water O$_2$ concentration for OB at the time sediments were cored, was 270 $\mu$mol L$^{-1}$. In the Ross Sea,

Marsay *et al*. (2014) estimated spatially variable efflux spanning 0.028-8.2 $\mu$mol m$^{-2}$ d$^{-1}$ based on water column dFe profiles (Marsay et al., 2014). Abundant epibenthic fauna found within Andvord (Ziegler *et al.* 2017, 2020) would introduce oxygen to the upper few centimeters of the sediments through bioturbation and reduce the efflux of reduced metals (Severmann et al., 2010). Taylor *et al*. (2020) found the upper 5 centimeters of Andvord Bay sediments possessed a high mixing coefficient relative to open shelf stations and Palmer Deep. Additionally, high inventories of $^{210}$Pb relative to open shelf and Palmer

Deep stations indicate a high mixing coefficient for sediments between 7 and 22 cm depth on timescales of 100 years (Taylor, DeMaster, and Burdige 2020). The effect of this process is mixing of oxide- and organic carbon-rich surficial sediments further down in the core on short- to long-timescales. These flux estimates, together with solid phase speciation results, highlight the importance of rapid oxidation and precipitation occurring at the seawater interface, which effectively retain Fe as oxy-hydroxides within the sediments (Burdige and Komada, 2020; Laufer-Meiser et al., 2021). The Fe oxides

are enriched within the penetration depth of oxygen (~0.5 cm, Fig. S2 inset) and once bioturbated downward, could be a source of dFe following microbial cycling. Multiple local maxima of porewater dFe were observed deeper in the cores. While dissimilatory iron reduction (henceforth, DIR) would be a source for Fe, oxidation of Fe with bottom water O$_2$ and Mn(IV) are important sinks and exert a control on the dFe concentration of deep water masses. The deep inner basin water column samples had high dFe concentrations concomitant with high LpFe concentrations (Fig. 2, 6), suggesting some loss of

porewater dFe to the water column and rapid formation of authigenic Fe mineral particles. Therefore, the fluxes calculated



from porewater profiles upper limit estimates because they do not account for oxidative losses at the sediment-water interface (e.g., Burdige and Komada 2020).

Due to weak midwater circulation, low tidal energy, and stratification of the surface, a disconnect between deep water masses enriched in dFe and the surface of Andvord Bay persists during prolonged quiescent periods. For these reasons, we

believe most sedimentary-sourced Fe is restricted to deep water masses and therefore plays a minor role in dFe concentrations within the upper water column. There is potential, however, for re-suspension and entrainment of surface sediments where subglacial meltwater discharges at the grounding line. Due to the low inferred volume of discharge this is likely a small contribution to the total particulate mass within the plume.

The Mn:Fe ratio is a useful signature of the source of dissolved and particulate trace metals in Antarctica and has been

applied to the PAL LTER data set (Annett et al., 2017). Applying this same framework to our study, we find that water column dissolved trace metals are heavily influenced by surface glacial ice melt and subglacial meltwater, and to a lesser extent, sediment sources within the fjord, irrespective of season, depth, and meteoric water input (Fig. 7). Due to the shorter residence time of dFe relative to dMn (i.e., inorganic oxidation of Mn is $10^7$ times slower than Fe, Sherrell *et al.*, 2018), we would expect the porewater dissolved Mn:Fe ratio to tend towards higher values once exposed to the seawater oxidative

front. We therefore cannot rule out porewaters as a source of dMn to the water column. A similar process occurs within the plume, where the elevated dissolved Mn:Fe (0.65 mol mol$^{-1}$) relative to labile particulate Mn:Fe (0.024 mol mol$^{-1}$) shows the effect of rapid conversion of Fe to authigenic mineral particles. Although we do not have comparable measurements for sedimentary labile particulate Mn, based on labile particulate Mn:Fe, we find that the water column labile particulate Mn:Fe ratio is precisely the same ratio as particles found within the subglacial plume, again irrespective of when and where the

sample was taken (Fig. 8), suggesting plume particles remain suspended throughout the fjord water column.

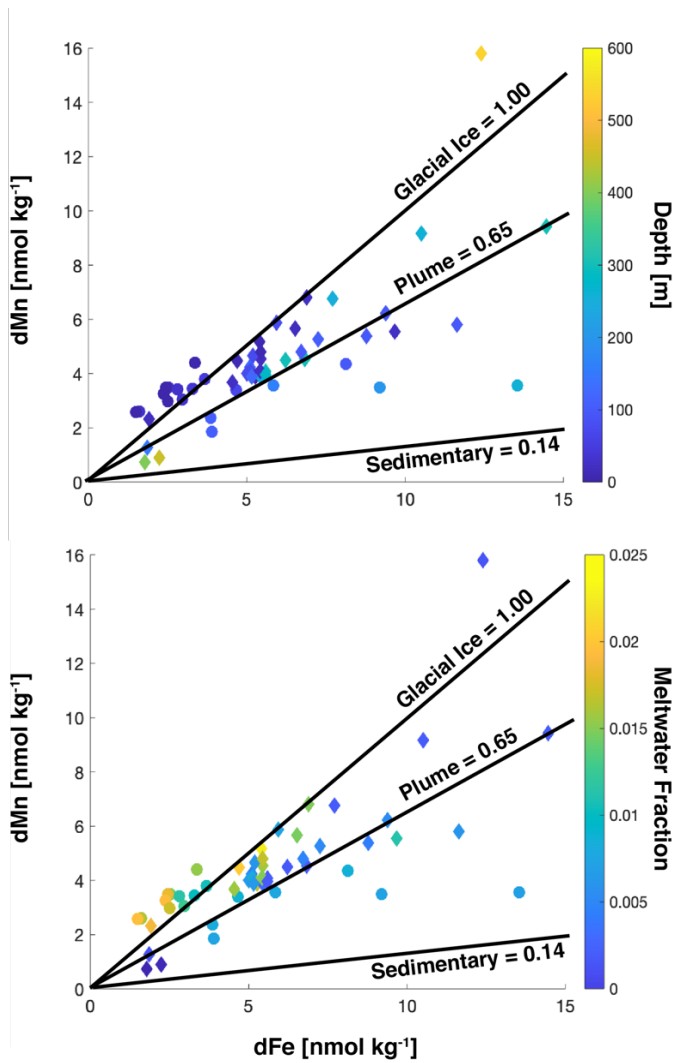

**Figure 7. Dissolved Fe and Mn plotted for water column samples. The colorbar shows depth (top panel) or meltwater fraction (bottom panel). For both panels, December 2015 cruise is indicated by filled circles and the April 2016 cruise is indicated by filled diamonds. The lines indicate the average Mn:Fe ratio for each candidate source.**

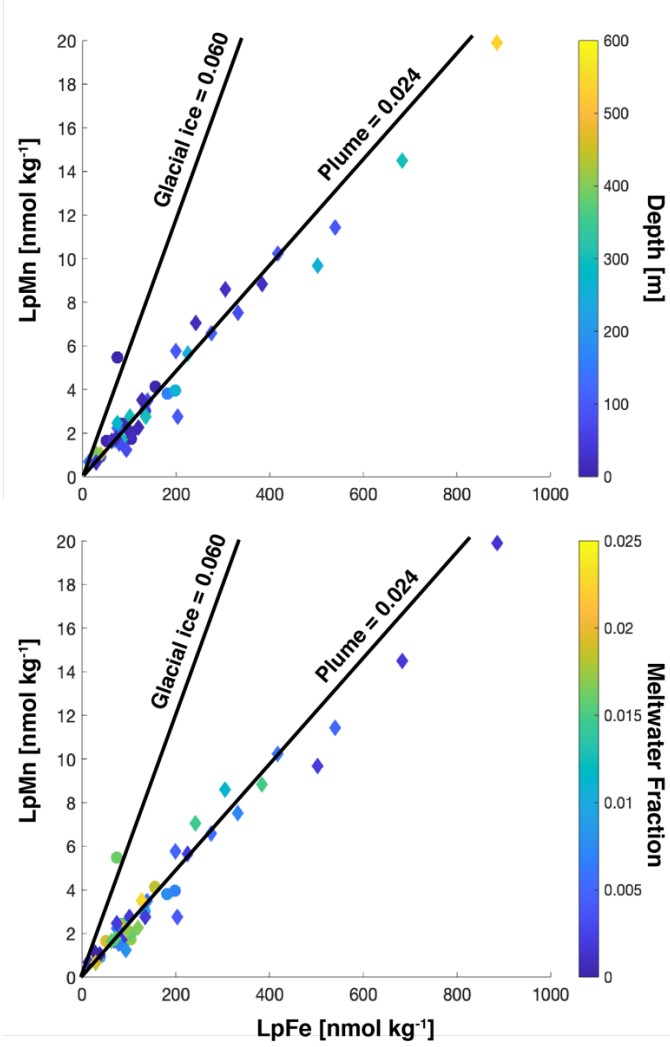

**Figure 8. Labile particulate Fe and Mn plotted for water column samples. The colorbar shows the influence of depth (top panel) or meltwater fraction (bottom panel). For both panels, December 2015 cruise is indicated by filled circles and the April 2016 cruise is indicated by filled diamonds. The lines indicate the average ratio of Mn:Fe determined from candidate sources.**

**4.5 Organic speciation of dissolved Fe**

It has been hypothesized that excess ligands ($eL = [L_t] – [dFe]$) increase the solubility of particulate Fe phases (Gledhill and Buck, 2012; Tagliabue et al., 2019; Thuróczy et al., 2011; Wagener et al., 2012). The persistence of exchangeable pools of dFe would therefore be controlled primarily by particle assemblage and organic ligand complements, where pFe dominates total Fe speciation. We observe a modest increase between late Spring and Fall in the relative contribution of dFe to total Fe (4% to 5% of TDFe, respectively), implying dFe is controlled by scaling closely to LpFe (Fig. 5g,h) since both pools have



large interseason differences. This corresponds to an increase in eL between seasons (average $2.1\pm1.3$ nM late Spring $n = 9$, $6.0\pm3.2$ nM Fall $n = 12$). The ligands are likely produced during microbial high-affinity uptake or remineralization processes following the termination of a bloom (Gledhill and Buck, 2012; Hogle et al., 2016). The only subsurface sample to contain strong Fe-binding ligands is the deep inner basin adjacent to Bagshawe Glacier (IBA), possibly indicating these ligands have

a sedimentary source. It appears, based on these results, ligands in Andvord Bay have the capacity to complex additional Fe input, as well as prevent significant loss due to scavenging (Thuróczy *et al.*, 2012). The nature of these ligands, taken together with the low concentration of dFe and abundance of LpFe within the plume, leads us to speculate that Fe minerals are the target for ligand-mediated mineral dissolution and perhaps microbial uptake, previously found to occur in deep-sea hydrothermal vent plumes (Li et al., 2014).

While we observe a seasonal increase in the excess ligand concentration, there is no significant change in the ratio of $L_t$:dFe (late Spring $1.8\pm0.5$, Fall $2.0\pm0.7$). In the Amundsen sector, Thuróczy *et al.* (2012) found waters heavily influenced by the Pine Island Glacier to have $L_t$:dFe ratios <2.5 throughout the water column, with relatively weaker ligands compared with those found in the highly productive surface waters of the polynya. We too identify weaker Fe-binding ligands associated with the glaciers, and only at MB and Sill 3 did we observe elevated $L_t$:dFe (3.13, and 2.99 respectively, in the Fall). In the

coastal zone of a remote island in the Bransfield Strait, an excess of strong Fe-binding ligands was observed, hypothesized to indicate Fe-limiting conditions (Buck et al., 2010). Temperature and salinity profiles show a strong signature of Bransfield Strait water within Andvord (Lundesgaard et al., 2020). The presence of excess strong Fe-binding ligands at IBA and S3 during the bloom onset also correspond to elevated $NO_3^-$:dFe (data not shown) above the threshold for potential Fe-limitation of coastal diatoms in the California Current transition zone ($\sim$10-12 µmol nmol$^{-1}$ King and Barbeau 2011). The presence of

strong Fe-binding ligands suggests an active microbial strategy in this coastal region to sequester additional Fe from particulate phases during the bloom initiation.

The intense seasonality in primary production and the presence of an undersaturated ligand pool could further increase the bioavailability of particles for downstream communities, where particles within the water column are rare. We calculated the capacity for the free Fe-binding ligands to bind Fe ($\alpha_{FeL'} = 1 + (eL \cdot K)$). Calculations of $\alpha_{FeL'}$ are included for each sample in

Table 2 as well as the Fe' inter-seasonal percent change for reoccupied stations ($R_{Fe'}$). We find the $\alpha_{FeL'}$ increased between late Spring and Fall at IBA, and Sill 4, while a decrease was found at IBB, Sill 3, and Gerlache Strait stations. While all reoccupied stations show an increase in the Fe' concentration ($R_{Fe'}$), the percent change is greatest where $\alpha_{FeL'}$ decreased in the Fall. Thus, the seasonal increase in Fe' reflects the increase in dFe concentrations as well as lower complexation coefficient of weaker Fe-ligand complexes, which contribute most to dFe speciation in the Fall and are associated with

surface waters adjacent to glaciers.

These first results of organic speciation of dFe in an Antarctic fjord highlight the importance of seasonal ligand sources in establishing the solubility of new Fe entering the coastal ocean. Seasonality in the ligand pool is not currently represented within SO biogeochemical models (Death *et al.*, 2014; Oliver *et al.,* 2019; St-Laurent *et al.*, 2019; Raiswell *et al.*, 2018; Person *et al.*, 2019). Ligand-mediated complexation has the potential to greatly expand the spatial extent in which



solubilization of particulate Fe occurs and could be critical for sustaining productivity over a larger geographical region
(Ardiningsih et al., 2020). Thus, the size, sinking rate, and composition of particles is critical to their lateral transport and
reactivity over time with excess ligands. Our understanding of how cryospheric Fe is transformed after entering the coastal
ocean is an important step towards understanding its impact on marine productivity and global biogeochemical cycles with
associated feedbacks on climate. For the marine Fe cycle, these geochemical transformations control the bioavailability of

Fe, while vertical advection and mixing supply this critical micronutrient to the surface ocean and the euphotic zone.

**4.6 Using dye experiments to explore Fe sources and export**

Rapid communication between the surface and subsurface water masses occurs during katabatic wind events. While mixing
can be more pronounced in the presence of icebergs, the large magnitude of vertical shear initializes an upwelling cell close
to the inner basins of the fjord. Using an idealized model of a fjord, Lundesgaard *et al.* (2018) found that katabatic winds

export the surface layer efficiently, for which several factors are considered important, including the wind velocity, elapsed
time of the event, and whether the wind is along-fjord versus off-axis. Within this idealized model of the fjord, the forcing
event leads to outcropping of deeper isohalines (up to 0.3 PSU greater) at the surface along the northern flank of the fjord,
corresponding to upwelling (see Figure 11 in Lundesgaard *et al.,* 2019). Wind-induced overturning circulation, along with
deepening of the mixed layer by up to 25 m, would increase surface dFe concentrations. These general model results showed

that wind forcing caused water at depths of 50-150 m to upwell rapidly (within 24 hours) near the glacier termini. This is an
important consequence we explore further in the highly-resolved model representation of the study region by Hahn-Woernle
*et al.* (2020).

The results of the dye experiments allow for the determination of fluxes, either prescribed (in the case of glacial meltwater)
or as a result of wind forcing. St. Laurent *et al.* applied similar methods in the Amundsen Sea with explicit coupling of sea

ice – ice sheet – ocean interactions (St-Laurent et al., 2017). In a more rigorous biogeochemical model, which included
ocean interactions with both sea ice and ice shelves, as well as parameterized Fe reactions, the productive waters in the
Amundsen Sea Polynya were supplied by an advected source of dFe from the "meltwater pump" and coastal currents, but
this model lacked explicit contributions of subglacial Fe (St-Laurent et al., 2019). These prior modeling results highlight the
importance of lateral exchange of surface water masses, and so the export of the surface out of the fjord mouth is explored in

the subsequent section.

**4.6.1 Surface meltwater sources and export**

The surface glacial meltwater flux was estimated in a previous section, assuming the meltwater produced by warm air
temperatures and solar irradiance is distributed evenly over the entire fjord area. We compared the observed and modeled
contributions of surface glacial meltwater and subsurface sources to the surface dFe inventory at two key stations, Sill 3 (S3)

and Gerlache Strait (GS). We assume that the concentration of dFe is a composite signature of three water masses (*surface
meltwater dye*, *subsurface dye*, *deep dye*), with varying relative contributions to the surface inventory. For example, in the





late Spring, at S3, we observed a MWf of 0.0155 and the surface concentration of dFe was 2.49 nM (Fig. 3). Assuming a meltwater end member concentration of 71.52 nM (Table 1), we find that glacial meltwater contributes 1.09 nM (44% of surface stock) to the surface inventory. In the Fall, the MWf at S3 increased to 0.0226 (Figure 3d), which corresponds to a

meltwater contribution of 1.59 nM dFe (35% of surface stock). In the GS, the same analysis reveals that in the late Spring, the surface had a MWf of 0.0193, which contributed 1.38 nM dFe (105% of surface stock) and in Fall had a MWf of 0.0169, or 1.21 nM dFe (24% of surface stock). It could be the case that the meltwater signature observed in the late Spring in the GS did not originate from Andvord Bay, and thus, might have a different dFe content, but the dearth of measurements of dFe in Antarctic glacial ice prevents us from testing this. These results, apart from those for GS in the late Spring, suggest that one

or several other sources contribute to the surface inventory of dFe, or, alternatively, that the glacial end member concentration is too low. Given that the MWf varies from 1-2.5% within Andvord Bay during the time of sampling, it is expected that the input of glacial meltwater throughout the melt season would supply some dFe to the surface.

When we examine the time series derived from the model, we find the model consistently underestimates the contribution of meltwater to the surface (Fig. S7). The MWf does not exceed 0.0013 at either S3 or GS stations, and its seasonal maximum

of 0.0046 is found at IBB in early February. Since processes like melting of drifting icebergs and sea ice cannot be captured in the model, the applied meltwater flux is based on a simplified representation of all new freshwater sources except for precipitation in Andvord Bay. These sources include, for example, surface runoff and local melt of glacial ice exposed to the atmosphere. The flux which best recreates observed salinity and temperature profiles in Andvord Bay was achieved by a meltwater input of 0.15 GT over 4 months (Hahn-Woernle et al., 2020).

The overall low modeled meltwater fraction is likely a consequence of multiple factors of which we discuss three. First, the meltwater was tracked only for the field season. The generally low salinity in the upper layer at the beginning of the season and the presence of meltwater dye at the end of the summer season (fjord average of 0.0003 MWf in upper 20m) suggested that meltwater can reside for multiple years in the fjord and cannot be fully captured by our meltwater dye. Second, local melt of glacial ice, e.g. floating icebergs, caused by a summertime surface heat flux, can have a strong impact on the MWf in

the surface layer and is likely to be underestimated and not well-represented with the parameterization of the modeled meltwater input. Third, only meltwater from the inner Andvord Bay is tracked and other sources are neglected. Based on other modeled meltwater dyes that track sources just outside Andvord Bay, the impact of the external sources is minor (maximum of 0.0003 MWf in early February) compared to the local sources, but they still contribute to the seasonal increase in MWf.

We extracted vertical profiles of MWf from the model at both stations and found that glacial meltwater originating from Bagshawe and Moser glaciers reaches maximum concentration during the summer bloom (late-January 2016) at Sill 3, relatively constrained to the upper 25m (Fig. S8b). In early February, when the bloom was terminated, glacial meltwater concentrations in the fjord decreased due to a weakening meltwater input and lateral dispersal. The weakening input is supposed to reflect the seasonal cycle of ice melting. Ocean circulation dispersed the meltwater into the Gerlache Strait, as

shown by a progressive increase in meltwater in the upper water column throughout the melt season (Fig. S8a). If the volume





flux of meltwater input is indeed correlated to the seasonal air temperature cycle, as it is parameterized in the model, the results in Fig. 3 would reaffirm that meltwater is an important control on the accumulation of phytoplankton biomass within Andvord Bay (Pan et al., 2020).

The effect of the wind in driving vertical fluxes will vary with wind direction and location within the fjord. The vertical
velocity is analyzed for the observation site at Sill 3 and in front of Bagshawe Glacier (IBA). The latter site is an example location for which katabatic winds are expected to lead to intensified upwelling and is also the location of the subsurface and deep dye experiments. Figure 9 (a) and (b) depict the relationship between the katabatic wind events and vertical velocities at 20m: landward-blowing wind generally leads to downwelling, while seaward-blowing katabatic wind leads to upwelling.

Based on observations of dFe from the late Spring prior to a wind event on December 11 ([dFe] at 20 m: 1.967 nM at S3,
2.006 nM at IBA), and the modeled maximum vertical velocities during the wind event ($2.094 \times 10^{-5}$ m s$^{-1}$ at S3, $5.083 \times 10^{-5}$ m s$^{-1}$ at IBA), we computed the upwelling flux of dFe into the surface (20m) at Sill 3 and IBA to be 3.54 µmol m$^{-2}$ d$^{-1}$ and 8.81 µmol m$^{-2}$ d$^{-1}$, respectively. These results shed light on the spatial heterogeneity of upwelling conditions within the fjord. Model results for Sill 3 are supported by late Spring observations of elevated dFe and low meltwater fraction at this station (Fig. 3). We argue that these punctuated periods of upwelling could be a substantial source of dFe to surface waters in
Andvord Bay. Further, this supply, together with the flux of glacial meltwater, provides dFe to fuel phytoplankton community growth.

The efficiency with which wind events export the fjord surface water is explored in the glacial meltwater dye experiment. To account for the changing amount of meltwater in the fjord, export across the fjord mouth in Fig. 9c is given as the percentage of the total amount of dye present within the fjord to resolve the effect of katabatic winds on dispersal dynamics of Fe-rich
sources. The meltwater dye experiences up to a 28-fold increased export into the Gerlache Strait during periods of strong along-fjord wind, primarily through the surface. To analyze the correlation between along-fjord wind velocity and the relative meltwater export, we first apply a 24-hr Gaussian filter to the relative export of glacial meltwater (Fig. 9), to exclude tidal signals. Applying the same filter to the wind time series, we find the wind and export data are positively correlated (r = 0.628). The correlation between export and along-fjord winds supports the results by Lundesgaard et al. (2019) who found
that katabatic winds control the export of fjord water. This has important implications for the dispersal of Fe-rich waters downstream, which eventually mix with Fe-poor waters located on the continental shelf (Annett *et al.*, 2017).


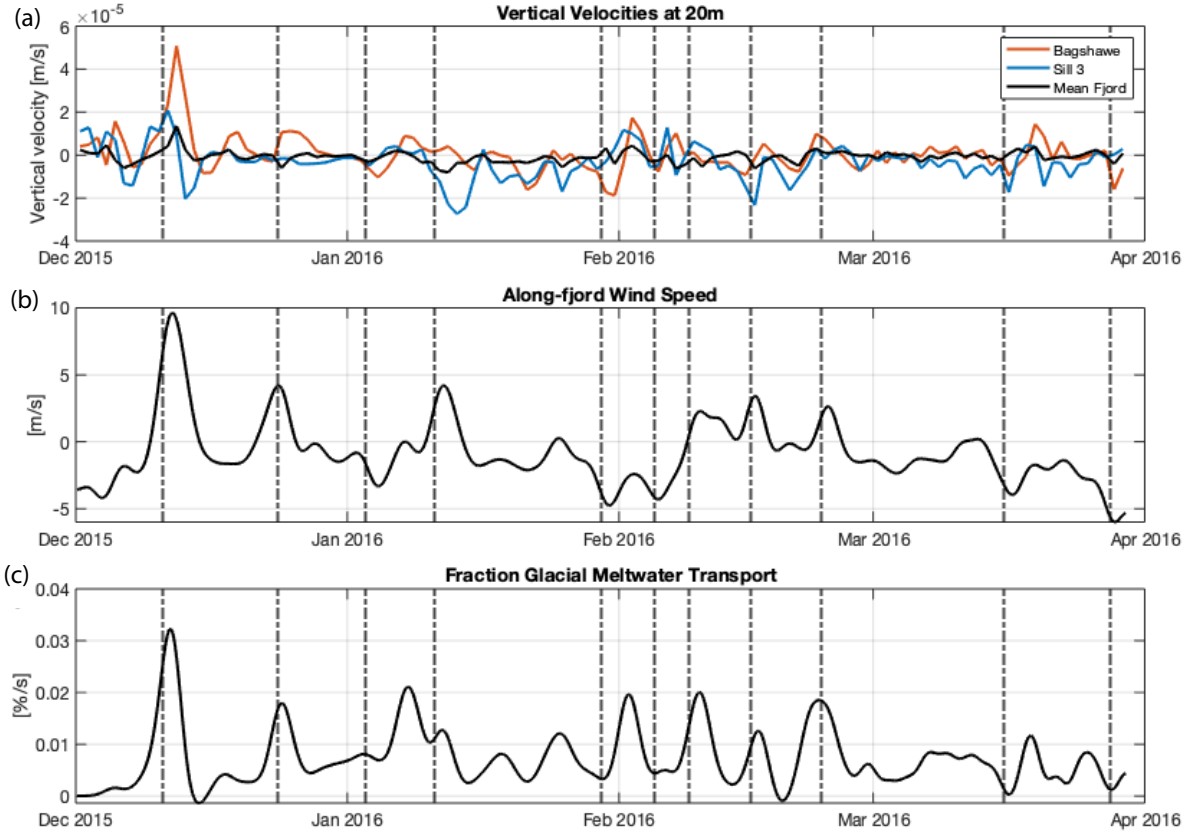

**Figure 9. (a)** Modeled vertical velocities at 20 m for the following locations: Bagshawe Glacier (IBA), Sill 3 and the fjord region average. 24 hr gaussian filter applied to time series of along-fjord wind velocity **(b)** and relative meltwater export out of the fjord **(c)**. Wind events exceeding an absolute velocity of 8 m s$^{-1}$ are indicated by vertical dashed lines. Wind speed data is based on bias-corrected RACMO model output for the center of the fjord, used to force the ROMS model. The transport of meltwater dye is shown relative to the total amount of meltwater dye within Andvord Bay to focus on the physical dynamics and not the changes in volume of dye present in the fjord.

### 4.6.2 Subsurface and deep sources and export

Periods of vertical mixing are shown to occur during katabatic wind events (Lundesgaard *et al*., 2019, 2020). This could be an important mechanism for supplying additional dFe from the subglacial plume to the surface within the fjord. Prior to the wind event on December 11, the subsurface dye increases gradually in the upper 20m (Fig. S6b). With the onset of the wind event, the vertical transport of the subsurface dye into the upper 20 m intensifies and reaches a maximum of 32.7 x 10$^3$ m$^3$ d$^{-1}$. In comparison, the deep dye does not enter the upper 20 m prior to the wind event and its maximum vertical transport is





only 4.2 x $10^3$ m$^3$ d$^{-1}$. It follows that katabatic wind events increase mixing in front of Bagshawe Glacier and have a particularly strong effect on water masses at intermediate depth. Assuming a mean concentration of 8.75 nM dFe for the subsurface plume (Table 1) and 8.68 nM dFe for deep (~300 m) IBA waters in the late Spring, these periods of vertical mixing correspond to dFe fluxes of up to 2.81 nmol dFe m$^{-2}$ d$^{-1}$ and 0.36 nmol m$^{-2}$ d$^{-1}$ (3.17 nmol m$^{-2}$ d$^{-1}$ combined) based on

the subsurface dye and deep dye, respectively. Following the katabatic wind event, which lasted approximately 11 days, model results show that 36% of the subsurface dye has shoaled above 75 m, with 10% of dye found within the surface layer (<20 m, Fig. S6b). Of the deep water dye, less than 1% is found within the surface layer. The behavior of the deep water masses contrasts with that of the subsurface water, which corroborates the geochemical data suggesting an insignificant contribution of deep water masses to the surface hydrography and thus, to surface dFe inventory. The vertical fluxes

estimated in this section are interpreted as a lower-bound for the contribution of the subsurface plume, since the modeled subglacial plume is a fixed volume, when in reality, subglacial meltwater might be supplied continually throughout the melt season. Compared to the flux of surface glacial meltwater input, and the flux due to subsurface and deep water mixing, the upwelling flux generated by wind events is the largest by an order of magnitude.

The quicker export of the subsurface dye, and therefore the low surface concentration, is mainly due to its proximity to the

ocean surface (Fig. S5b). The upper ocean is more subject to changes in the upper ocean dynamics and wind stress. In contrast, the deep dye is exported more slowly and is more continuously released into the Gerlache Strait (Fig. S5c). These modeling results provide evidence for the flushing of fjord water to the Gerlache Strait which coincides with periods of intensified winds. Thus, katabatic winds are important both for replenishing the surface Fe concentrations from the subglacial plume as well as exporting Fe-rich surface waters. It is reasonable to assume that in the absence of a strengthened

buoyancy-driven overturning circulation, sources from fjord sediments are limited in supplying the surface with dFe in Andvord Bay.

## 5 Conclusion: Andvord Bay as a source of Fe

We have shown that in the absence of buoyancy-driven upwelling, the interaction of the ice sheet, atmosphere, and surface ocean, is important for resupplying the surface waters with Fe throughout the summer season, leading to enhanced

productivity and sedimentation of carbon. Katabatic wind events result in pulsed export of the surface layer, while upwelling and vertical mixing entrains subglacial plume water in the inner fjord. Observed surface concentrations of dFe in Fall lend support to the modeled dynamics (see Fig. 3). We summarize the findings of this study in a conceptual diagram showing important seasonal sources of Fe during the growth and melt season (Fig. 10). We highlight important processes in the diagram using circled number notation. We found ocean temperatures are cold ① and do not melt the fronts of glaciers, but

warm summer atmospheric temperatures contribute to the surface melting of glacial ice ②. Variability in dissolved and particulate concentrations in glacial ice produces large uncertainties in the calculated flux. The speciation of Fe within glacial ice is mostly accounted for by refractory Fe-bearing particles ③. Only a fraction may be stabilized by excess organic



ligands. Another source is fjord sediments ④, though there is considerable uncertainty shown in the magnitude of this flux because evidence indicates that a significant fraction of porewater Fe rapidly precipitates at the oxidative front forming a rich surface layer of Fe oxyhydroxides ⑤. Intense bioturbation of fjord sediments mixes the surface sediments downwards fueling dissimilatory reduction processes. The dFe that escapes this sink enriches deep waters within the fjord. Small amounts of subglacial meltwater discharge enter the ocean and form turbid subsurface plumes ⑥. Within the plume, speciation is dominated by high concentrations of labile authigenic Fe-bearing particles that can be solubilized by Fe-binding organic ligands ⑦. Seaward-blowing katabatic winds ⑧ occur episodically and cause upwelling and vertical mixing supplying additional Fe to the surface phytoplankton community. These intense energetic periods facilitate the dispersion and export of surface Fe and meltwater away from the fjord where it is advected downstream in the Gerlache Strait ⑨.

Given that the west Antarctic Peninsula hosts the greatest number of glaciomarine fjords on the continent, and multiple katabatic wind events occur throughout the year, single wind events can play a crucial role for the export of Fe. The modeled export of meltwater integrated over the week after the wind event on December 11 is $38 \times 10^7$ m$^3$, which is about 43% of the meltwater input during the same time. For comparison, during the following week, with relatively calm wind conditions, only 20% of the meltwater input is exported. We estimate the Fe export to be 272 mol dFe week$^{-1}$ for this event. However, the warming climate may lessen the likelihood for pulsed export of meltwater-derived Fe by intensifying coastal currents due to declines in sea ice (Moffat et al., 2008), and reduced surface cooling, decreasing the velocity and frequency of katabatic winds over the west Antarctic Ice Sheet (Bintanja et al., 2014).




**Figure 10. Conceptual diagram showing the important seasonal sources of new Fe during the growth and melt season. The red arrows indicate the major fluxes (in [$\mu$M m$^{-2}$ d$^{-1}$]), with the size ranges showing the uncertainty in the measurement – some fluxes are difficult to quantify. These fluxes also vary from season to season and from location to location and may even be going through long-term changes due to human influences, such as climate change, though this is not shown here. The small arrows show internal**
**transformations of Fe, which play an important role in the supply of Fe to phytoplankton. See text for a description of important processes highlighted by circled numbers.**

The large variability in inferred dFe content of glacial meltwaters along the WAP (Annett et al., 2017) means that supply likely depends on fjord-specific processes and future changes in ice volume. Advected sources of dFe remain the largest
contribution (~50%) to the inventory on the productive continental shelves (De Jong et al., 2015). Therefore, we believe that a latitudinal assessment of WAP fjords could begin to address variable responses to ocean and atmospheric forcing in these



productive ecosystems. Indeed, less than 160 km to the south of Andvord Bay, observations of warm modified UCDW intrusions and an invigorated "meltwater pump" present an alternative mechanism for sustaining local primary production (Cape et al., 2019).

The scope of our results should be highlighted. If we assume Andvord Bay is representative of a typical cold-based fjord, and similarly, Barilari Bay is representative of a warm-based fjord (6% MWf at surface, Cape *et al.*, 2019) then we can estimate the glacial meltwater export resulting from a single wind event for the entire western coast of the WAP (see *Appendix A*). A total of $3.6 \times 10^{10}$ m$^3$ (36 km$^3$) of surface glacial meltwater is exported seaward, which corresponds to $2.0 \times 10^6$ mol dFe. A modelling study estimated a total meltwater discharge for all of Antarctica to be 32.5 – 97.5 km$^3$ yr$^{-1}$ (Pattyn

2010). Thus, katabatic winds are highly efficient at delivering surface meltwater produced near the coast to the continental shelves. However, this is small compared to the total basal melt production rate due to warm ocean temperatures for the largest ice shelves. Using highly accurate remote sensing topographic measurements Adusumilli *et al.* (2020) found that the major ice sheets have a steady-state meltwater production value of 1100±60 km$^3$ yr$^{-1}$. In a different modeling study, it was estimated 300 – 800 km$^3$ yr$^{-1}$ enters the SO accounting for observed trends in SO sea surface temperature, sea ice expansion,

and sea surface height (Rye *et al.,* 2020). The WAP feeds most directly into the Antarctic Circumpolar Current (ACC), which advects modified coastal waters downstream to the productive Scotia Sea region, potentially magnifying the ecological impact of WAP fjord meltwater production. As the next wave of ocean biogeochemical models incorporate processes at the ice-ocean interface, better predictions of Fe supply to the ACC will be made.

In Andvord Bay, primary production will be sensitive to future changes in subglacial discharge as Antarctic glaciers continue

to melt in response to oceanic and atmospheric warming (Smith et al., 2020). A greater flux of sediment is expected to be released into the fjord, reducing light quality for primary producers, as part of a natural tidewater glacier cycle (Brinkerhoff et al., 2017). A key question outside the scope of this research is how the quantity and quality of Fe-binding ligands will change in the future. To a first approximation, decreased ligand concentrations associated with the phytoplankton bloom are expected to reduce efficacy of solubilization of particulate Fe and natural fertilization downstream resulting from this *leaky*

fjord. This climatic trend is not yet realized within Andvord Bay (Eidam et al., 2019), but is expected to decrease dFe export through increased scavenging and sedimentation, further resembling high-Arctic and temperate fjords (Hopwood *et al.*, 2016).

**Appendices**

**Appendix A: Estimating total meltwater export from WAP fjords**

In order to estimate the meltwater export resulting from a single katabatic wind event along the WAP, we first identify two fjord types: 1) fjords where waters are below the freezing temperature (cold-based); and 2) fjords where intrusions of modified UCDW reach the glacier terminus and cause melting. This distinction leads to different MWf production rates. We use data collected from Andvord Bay as a basis for the amount of export occurring in cold-based fjords.





In this instance, a maximum MWf or 0.025 was observed, which corresponded to an export of $38 \times 10^7$ m$^3$ glacial meltwater

and is based on the glacial meltwater dye export across the mouth of Andvord Bay integrated over the duration of a week-long katabatic wind event.

Meltwater runoff from glaciers due to warm atmospheric temperatures is parameterized as a function of number of days above a temperature threshold (Smith *et al.,* 1998). The area of the glacier in contact with the atmosphere predicts how much meltwater is generated. We use this simple relationship with surface area and relate it to the MWf we observe,

allowing us to estimate the fractional contribution from each glacier in Andvord Bay. As an example, Bagshawe Glacier has an area of 250 km$^2$, which is 48% of the total glacier area for this fjord, and so would be responsible for producing 48% of the surface glacial meltwater ($\sim 18.4 \times 10^7$ m$^3$). By dividing the total surface glacial meltwater export for a single katabatic wind event by the total area of glaciers in Andvord Bay, we calculate the export rate of meltwater in Andvord Bay glaciers to be $7.4 \times 10^5$ m$^3$ km$^{-2}$ assuming glaciers have an equal rate of meltwater production per unit area. We use this rate as

representative for cold-based type glaciers.

Since warm atmospheric temperatures in contact with the glacier surface cause production of meltwater, which enters the ocean as surface runoff, this seems a reasonable assumption. Additionally, intrusions of modified UCDW can reach the glacier terminus, causing slightly higher fractions of meltwater at the surface ($\sim 0.06$ in Barilari Bay). Our general model results showed exchange with water outside of the fjord occurred during katabatic wind events, including inflow of

water masses at depth located from outside of the fjord. Thus, these events are likely to enhance delivery of modified UCDW to the glacier terminus (Jackson *et al*., 2014). We scale the meltwater export to the meltwater fraction since both Barilari and Andvord Bays had similar mixed layer depths. Also, $\sim 40\%$ export of meltwater during katabatic wind events in our model is reasonable compared to estimates for Arctic fjords (10-50%, Jackson *et al*., 2014). Based on the area of glaciers in Barilari, we calculate an export rate of meltwater for representative warm-based glaciers to be $10.2 \times 10^5$ m$^3$ km$^{-2}$. We extrapolate

these rough estimates for all glaciers on the western coast of the WAP identified by Cook et al. (2016). All glaciers to the south of Andvord Bay are considered warm-based, while those to the north are cold-based (Fig. A1). The area of each of the glaciers used here is published in Cook *et al.* (2016).





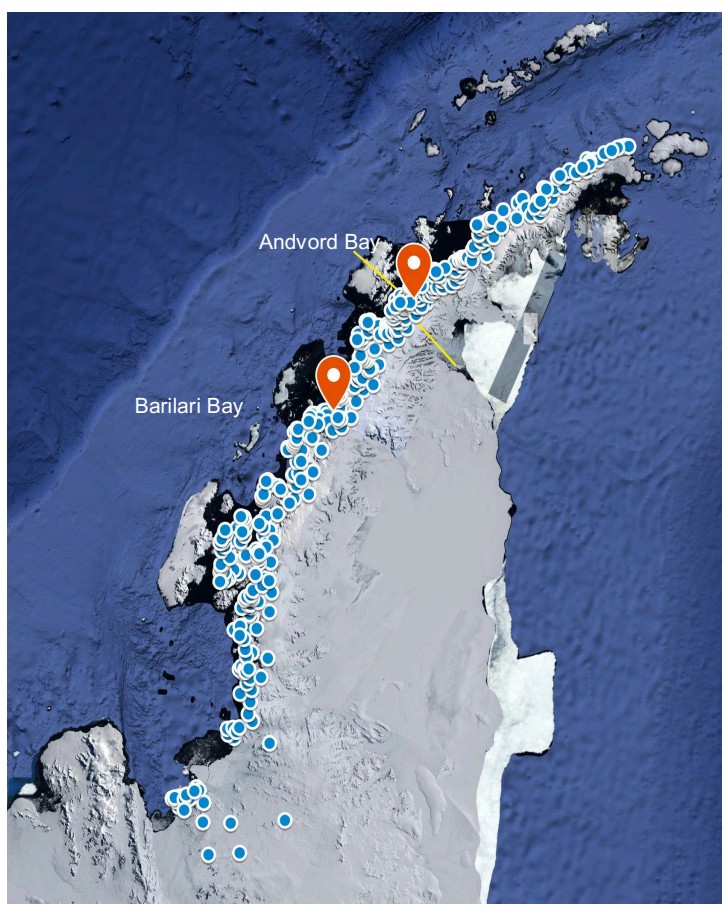

**Figure A1. Map showing all 432 glaciers (blue dots) located on the western coast of the WAP (from Cook *et al.*, 2016). The yellow**
**line indicates the region of convergence of two intermediate water masses; cold Weddell Water to the north and warm modified**
**UCDW to the south. Image was produced using © Google Maps, 10 January 2021.**

Summing the entire volume export of surface glacial meltwater, we find that if all surface waters along the western

coast of the WAP experienced a single katabatic wind event, reminiscent of the one recorded in Andvord Bay, a total of 3.6

x $10^{10}$ m$^3$ (36 km$^3$) of surface glacial meltwater is exported towards the continental shelf (5 km$^3$ from cold-based glaciers; 31

km$^3$ from warm-based glaciers). This latitudinal difference is consistent with greater meltwater fractions found on the

continental shelf in the southern lines of the PAL LTER grid (Annett et al. 2017). Based on a recent compilation of TDFe

content in icebergs from Antarctica (Hopwood et al. 2019), and including two measurements from our study, we use a

median concentration of 544 nM (*n* = 57). We then assume a conservative solubility of 10% of TDFe as the dissolved phase,

which yields a dFe content of glacial meltwater to be 54.4 nM. This is close to our average dFe measured for three glacial ice

pieces in this study (~71 nM). We estimate a single wind event lasting one week on the western coast of the WAP

corresponds to an export of 2.0 x $10^6$ mol dFe.


We realize this analysis does not take in to account the impact of shallow sills in fjords that might be important for restricting UCDW from entering the fjord mouth and interacting with glaciers. Invigorated upwelling due to buoyant plumes originating at the glacier face is expected to have a positive feedback on the melting of the glacier terminus by increasing the delivery of modified UCDW to glaciers and enhancing melt (Cape *et al.,* 2019). This may be driven by warm ocean temperatures, directly melting the face of the glaciers, or atmospheric warming could increase drainage of surface melt to the base of the glacier, resulting in subglacial discharge and buoyant plumes driving circulation. Directionality of the katabatic winds is an important parameter for wind forcing in fjords surrounded by steep topographic features (Lundesgaard *et al.,* 2018). We have explored the possibility when one katabatic wind event per year occurs in the along-fjord direction (seaward) for the entire western coast of the WAP. These mechanisms are fjord specific and deserve further attention due to the complex interactions between the ice, ocean, and atmosphere. We also concede that areal extent of glaciers may not be the most representative measure for meltwater production, when in fact glacier flow velocities might better correlate with meltwater production rates, and thus, meltwater export rates. However, the interplay between surface melt and the subglacial hydrological system, and thus flow rates could mean this is a sufficient, albeit rough assumption. Finally, large uncertainties exist for the average glacial ice content of dFe and the degree to which TDFe may be solubilized and made bioavailable. This analysis does not take into account the large quantities of solid ice (i.e., icebergs) exported via this mechanism.

**Data availability**

All CTD data from this study is available at U.S. Antarctic Program (USAP) Data Center: Vernet, M. et al. (2019) "FjordEco Phytoplankton Ecology Dataset in Andvord Bay" U.S. Antarctic Program (USAP) Data Center. doi: https://doi.org/10.15784/601158.

**Author contributions**

K.F. designed the study, conducted the analyses and led the writing of the manuscript. L.H. performed numerical dye simulations. R.S., J.R., and K.B. assisted in the preparation of multi-elemental analyses. D.B. provided data for sediment core analyses. L.H., R.S., D.B., M.V., and K.A.B. were involved in discussing the results and their implications, and contributed to the drafting of the manuscript.

**Competing interests**

The authors declare that they have no conflict of interest.





## Acknowledgments

The authors would like to thank all participating principal investigators and their affiliates during the NSF FjordEco project (PLR -1443705). Dr. Lauren Manck (University of Montana) assisted with sampling efforts during *NBP1603*. We would like to thank the captain and crew of R/V Laurence M. Gould and RVIB Nathaniel B. Palmer and United States Antarctic Program contractors. We also thank Dr. Brian Powell (University of Hawai`i, Manoa) for helpful comments and for providing resources for the modeling effort.

## Financial support


This research has been supported by the National Science Foundation (grant no. PLR -1443705). K.F. was supported by an NSF GRF (NSF 15-597).

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
