# Peer review of "Seasonal dispersal of fjord meltwaters as an important source of iron and manganese to coastal Antarctic phytoplankton"

_Biogeosciences, 2021_

## Author Comment (AC1)

Forsch et al., provide an unusually comprehensive study surveying the distribution of Fe and Mn in an Antarctic fjord, which has been the site of ongoing work by the FjordEco project. In addition to conducting profiles of the water column, the authors report sedimentary work, some analysis of ice samples, ligands, and some regional model work to comment in more detail on sources of Fe/Mn. The Fe:Mn ratio is also used to provide insight into the relative importance of different trace metal sources. Overall this is quite a novel study, there are few studies reporting depth profiles of these elements, which limit primary production across much of the Southern Ocean, in Antarctic coastal areas. The combination of data makes the study unique and a valuable addition to the literature. The visiting of the same site in two seasons is particularly valuable for an Antarctic fieldsite.

As written at present it is however quite long and in places I think more suitable for a Marine Chemistry readership, I think the text could be shortened a little. Some of the extrapolations from model work and calculations based on only a few melted ice samples could be trimmed a fair bit. This would strengthen the scientific arguments presented, cut out the parts of the discussion where large uncertainties remain and not much informative can be said, and make the text more readable.

This is a minor critique however, and overall the text is a strong addition to the field.

R: We thank Reviewer #1 for their detailed review of all aspects of the text. We recognize that at present the text is long and this required considerable effort on the part of the reviewer. The text will be shortened considerably by shortening and moving most of the modelling discussion to supplemental text. We will limit the discussion of extrapolative aspects where few measurements or large uncertainties prevent concrete conclusions. We address each comment by line using boldface formatted text.

Comments/corrections by line

Title: Why not 'Fe and Mn'?

R: We chose to focus this work on iron, making use of manganese as a proposed tracer for different iron sources. However, as both Reviewers indicated, Mn dispersal should be explicitly addressed due to the importance of both micronutrients for primary production in Antarctica. We will increase the focus on Mn as a micronutrient in the revised version.

15 'of bioavailable Fe' why not just 'Fe'?

R: Changed to just 'Fe'.

**23 Ocean or atmospheric temperature?**

**R: We have specified this as atmospheric temperature.**

28-29 This isn't strictly speaking correct, there isn't a simple relationship between increasing phytoplankton productivity and increasing carbon export because carbon export efficiency varies markedly between regimes (Henson et al., 2019), it would be more precise to say that Fe addition to Fe-limited regions increases primary production and potentially carbon export.

**R: We have changed this statement to be more precise: "...and when enhanced in Fe-limited regions of the ocean, naturally or artificially, primary production increases and potentially carbon export."**

45 'and reduced macronutrient supply'. It isn't turbidity that does this on broad scales – although there may be a very small phosphate sink onto Fe-rich particles – it's strong stratification that leads to very low productivity in some Arctic fjords (Holding et al., 2019). Rephrase.

**R: Rephrased.**

48 There's also the question of chemistry and factors that control Fe stability which you develop later. If discharge increases into a region which already has nM concentrations of dFe, is it possible to increase dFe and lateral dFe fluxes further? (Lippiatt et al., 2010) among other more recent references hints that dFe may be saturated in some near-shore, in this case Alaskan, regions which implies that increasing deposition of dFe or labile particles inshore wouldn't changes lateral dFe fluxes. More recent GEOTRACES work also comments on the competition between oxyhydroxide surfaces and ligands to bind Fe such that in these high turbidity environments undersaturation may be driven by increasing particle (and Fe) loads (Ardiningsih et al., 2021).

R: We show robustly that organic Fe-binding ligands are undersaturated everywhere in the fjord system. It is possible that this is a result of high concentrations of particle-associated ligands, which are sufficiently high in this environment to compete with the organic ligand pool and remove dFe. However, we imply multiple ligand sources which could easily obscure our ability to test this important hypothesis. The sentence in question does include the term "scavenging", which encompasses a number of potential chemical processes.

54 (I'm not a glaciologist) Cold-based - is this the correct term? My understanding from the literature was that cold-based and warm-based terms are not used to refer to submarine ice, only to land-based ice (e.g. see entry in Encyclopedia of Snow, Ice and

Glaciers). I think a different term is required when comparing submerged ice faces that are/are not subject to submarine melt.

**R: We have changed our verbiage to reflect more accurately the marine focus. Cold-water and warm-water are now used throughout the text.**

59 Note sure what 'minimal alteration' means in this context? You mean elsewhere there is also melting of the ice terminus – but I thought this was usually a very minor component of total freshwater discharge even in warmer catchments so I'm not sure it's much of a critical difference?

R: We have chosen to make this distinction because of the non-linear effects of mixing and dilution which occur at increasing freshwater input. Since there exists a small fraction of meltwater in the system as a whole, mixing ratios and vertical velocities are small compared to other catchments that have either larger subglacial or basal meltwater fluxes.

For clarity, we have changed 'minimal alteration' to be 'minimal dilution of seawater.'

61 anoxia- is this always, or only sometimes the case?

R: It is possible that anoxia does not develop as frequently as literature suggests, and could even demonstrate seasonality depending on the occurrence of flushing events, or exchange with the coastal ocean in subglacial cavities. We have changed the language to reflect this uncertainty.

78 "prior to significant glacier retreat" does not read well without a sentence explaining that this is(?) forecast/anticipated at this location. Also, I assume, you should specific prior to retreat associated with recent climate change?

R: Pritchard and Vaughan (2007) show widespread increases in the mean flow rate of marine terminating glaciers in the northern Antarctic Peninsula (increase of 12% from 1993 to 2003), coincident with frontal retreat over the period of study (1965 – 2004). As rates of summer warming increase in this region as a result of climate change, these trends are expected to continue into the future. We will cite this reference here and specify the retreat is associated with recent climate change.

88 Two two cruises

**R: Resolved.**

91 Neko Harbor – I would not know where this is without a dot on the chart or a lat/long

**R: Neko Harbor station will be added to the map in Fig. 1.**

Fig 1. I struggled to read the text on the right hand side of this figure, maybe improve the contrast of strengthen the outline

**R: We will change the text formatting to improve the contrast.**

106  $N_2$  and it would be better to state the specific grade e.g. 99.99%.

**R: Resolved.**

115 and filtered prior to analysis, repeated

**R: Resolved.**

120 Is there a specific reason for mentioning GP16, reads a little odd?

R: There was no intent for including GP16 besides to support the application of methods used in the current study. We have included a more general citation for GEOTRACES methods (Cutter and Bruland, 2012).

133 Q- is quartz distilled? Selectively means you varied the concentration factor?

**R: The grade is Suprapur. Selectively means we only pre-concentrate iron and not the high salt matrix. The concentration factor was not varied. We have changed this sentence to instead say "pre-concentrations and matrix removal."**

138 It would be better to say what this was e.g.. x nM dFe Pacific surface seawater

**R: Changed.**

144 Define LOD

**R: LOD is now explicitly defined as 3x the standard deviation of the blank.**

167 0.2% is ambiguous, M concentrations would be better

**R: We have instead stated "0.2% v/v."**

168 solution of solution

**R: Resolved.**

181 If these are average values, why ~? Surely they are exact. Are they meant to imply the gradient is subject to high uncertainty? If so, maybe show the uncertainty.

**R: We removed the tilde as these are calculated values.**

188 Reference format

**R: We have revised the reference formatting.**

207 ambiguous: ,for deeper layers, is 84.6 m and the minimum thickness, for surface layers, is 0.5 m (?)

**R: The modeled water column is composed of 25 depth layers, which follow the bathymetry such that the deepest layer can be up to 84.6 m thick, while the surface layers may be as small as 0.5 m thick.**

208 It might help some readers not familiar with the region if the Straits were labeled on Fig. 1

**R: Straits are now labeled on the left side map.**

214 'captured' Maybe 'represented'

**R: Changed.**

215 'These new freshwater sources include also surface runoff and local melt of glacial ice' repetition

R: We will revise this section by combining sentences in lines 213 and 216: "Processes like melting of icebergs and floating sea ice are not modeled directly, therefore such local freshwater sources are captured in a surface intensified meltwater input applied along the glacial fronts (for further details see Hahn-Woernle et al. 2020)."

232-235 I don't think you need this.

**R: Topic sentences are removed.**

240 What increased Si are you referring to? The innermost station looks depleted and the water column from 0-10 km on fig. looks like high Si-high NO3. A 'new' Si signal

would, I assume, show an excess of Si over NO3 if originating from bedrock related sources– you can test this by plotting Si\* and considering the origin of the watermasses. Is this Si high relative to NO3, and if not is the different macronutrient concentration in this watermass related to seasonal inflow/outflow and the sluggish turnover beneath the sill?

R: Plotted below are Si:N [mol:mol] ratios for late Spring (top panel) and Fall (bottom panel). These panels show where Si is enriched or depleted with respect to N, or in the case for the upper 50 m, where nitrate has been preferentially depleted. We excluded the upper 50 m to not confound our interpretation with biological drawdown. We notice that Si concentrations are enriched, relative to N, in the inner fjord (section distance = 0 km) and near-surface in the late Spring. The near-surface signature also shows high Si:N ratios for both seasons, and probably results from a combination of downward mixing of low N water following the phytoplankton blooms and/or due to the presence of glacial meltwater, enriched in Si and not N (discussed in new publication and references therein: Krisch et al. 2021). Indeed, Si:N is highly correlated with meltwater fraction in Andvord Bay (not shown). We note that in the Fall (bottom panel), the Si:N ratio everywhere is elevated relative to the late Spring (top panel).

We therefore suggest that elevated concentrations of Si in the late Spring are driven by fjord specific processes (i.e. meltwater/sedimentary input) and not a result of distinct water masses. The origin of the high Si:N signal in deep water masses outside of the fjord in the Spring (bottom panel; section distances > 20 km) is from Bransfield Strait water mass, which does enter the mouth of the fjord during katabatic flushing events (Lundesgaard et al. 2018). We will revise the text to indicate that the "Increased Si concentrations within the inner fjord" with respect to nitrate concentrations, are "driven by sedimentary processes, or weathering of the bedrock" since Si:N is highly correlated with MWf.

Fig 2 It would be useful to see where these stations are in order for the reader to be easily able to interpret the trends. Can you, for example, overlay the transect line on figure 1. What drives the 1 station with really low chl a in Dec, is this real, it looks suspicious/erroneous as plotted?

R: The transect line is plotted as a dashed yellow line in Figure 1, however, bounds for which stations are included will be mapped in the future version of the figure. The low Chla is anomalous and was located in a low productivity station located outside of the fjord mouth. We can remove this station from the section plot.

277 You can presumably calculate the upper limit though, if you assume all freshwater required to balance MWf came from this one point source (obviously thereby easily an upper limit), you would get a discharge of <1 m3 s-1 (correct?) which means this is unlikely to be driving considerable circulation.

R: The estimated volume flux of ~0.5 m3 s-1 leads to a weak buoyancy forcing and leads to a proportionally small entrainment rate (Ø. Lundesgaard PhD Thesis, University of Hawaii at Manoa). We will revise this sentence to include an estimate of discharge and a statement that this is not driving circulation.

299 Is this decrease significant?

R: The decrease in the mean is not significant. This does not change any interpretations in the rest of the manuscript. We will remove: "but a seasonal decrease in concentration was observed" and replace with, "and similar mean concentrations were observed for both seasons".

325 But you measured TdFe? So why not just 1 sentence comparing TdFe values?

**R: Agreed. We will instead compare TDFe with the Ryder Bay data set. However, we will keep the sentence that TDFe and LpFe are valid comparisons at these two sites since concentrations of TDFe >> dFe.**

360 Yes, these seem extremely high, I'm not sure if many prior values are published, the only ones I'm aware of are Al in (Menzel Barraqueta et al., 2018) who report much lower levels for Al. I note however that the authors' elemental ratios do seem sensible, so it looks like it just happened to be the case that the ice collected had a high sediment load, do you know (roughly?) what this was?

R: The anomaly is a result of targeted sampling of an iceberg with a high particle load, with dark layers of embedded sediment. The choice of regional crustal elemental ratios is open to debate, as we know that the Peninsula region has widespread volcanism and metamorphism (Jordan et al. 2020) and thus, might have different ratios than typical continental crust (ie basalt). The crustal component may be re-estimated in the subsequent version. This will change our results by increasing the crustal component of Fe and Mn, since these metals are enriched (relative to Al) in basaltic/andesitic crusts.

Table 1 The significant figures here could be reduced a bit, it doesn't really make sense to report decimal places for the high concentrations as written for example.

**R: Agreed. Significant figures are reduced for high concentration (glacial ice) samples.**

374 Details of statistical test

**R: No statistical test. The means are indistinguishable.**

Table 2 Check sig. figs. A few values are either rounded or missing .0

**R: Resolved.**

452 value\_unit consistency

**R: Resolved.**

478 "we note that the icebergs within Andvord were predominantly "clean" ice" How do you know this? And does this mean you intentionally sampled some ice which was sediment-rich when selecting the ice endmember samples?

R: This was based on visual inspection of icebergs during the field operations, but is also supported by glacial cameras, which monitored ice conditions within the fjord year-round. Two samples were selected as medium-to-high sediment load endmember samples to capture the variability present within the fjord. It is important to note that despite its relative smaller contribution to solid ice flux, "dirty" ice would contribute potentially as much or more dFe to the surface ocean upon melting.

493 "Average" means a mean? (I think in this context it's important to stress the mean/median values are likely very different)

R: The value reflects a mean, while the median would be resistant to outliers and be overall much lower since 90% of glacial ice values would fall below the arithmetic mean (Hopwood et al. 2019). We are limited to using the LpFe mean of two glacial ice samples to compare to the largest compilation, since there are few measurements of dFe in glacial ice from Antarctica and it is not clear how dFe scales with TDFe in glacial ice. We will add a sentence about how the median value would be expected to be lower than this mean (closer to Glacial Ice 3 and 4).

501 'might' can probably be removed here, it's obvious from your data scavenging does occur, as it does everywhere else.

**R: Agreed, so we have removed "might."**

515 "It seems reasonable..." repetition of the last few sentences

**R: Resolved.**

518 "(82-86% of TpFe, 61-64% of TpMn)" It's not clear at a glance what measurement the % refers to as a fraction of TpFe/Mn

R: These percentages reflect our estimate of contributions from crustal material to the total particulate trace metal concentrations. We make use of the ratio of

elements found in the upper continental crust. We calculate the contribution of crustal material to total particulate trace metals by the following equation:

**% crustal = ([TpAl]\*Me:Alcrustal)/[TpMe]**

This will be clarified in the subsequent version of the manuscript.

526 delete 'as'

**R: Resolved.**

527 As above, is it generally correct to state the subglacial environment was certainly anoxic, or does this vary with location? Do you have evidence specifically in this region that it is anoxic?

R: We do not have direct evidence that the subglacial environment is anoxic. This is inferred due to the high concentrations of labile particulate metals observed, which would form rapidly upon mixing of reduced species with oxygenic seawater. There is strong evidence for anoxia in general in Antarctica, as opposed to Greenland, where surface melt enters the subglacial system via moulins.

533 I think you need to state what this (8 nM dFe) is 'low' compared to (subglacial dFe?), in a marine context it's very high

**R: We have revised the sentence to read as "low compared to subglacial fluids in contact with bedrock."**

534 You need to state here what you're assuming the freshwater content is, basal ice? These sentences I think are speculative, if you look at any freshwater studies trying to quantify dFe (granted, there are no extensive surveys of freshwater dFe in runoff along the WAP that I am aware of, or similarly for subglacial discharge) the range is huge, so an obvious caveat is that you don't really know exactly what the freshwater concentration corresponding to these marine values is/was – and even if you did, it would likely vary so much in time and space that this variation would preclude any direct calculations concerning the exact weighted concentration most appropriate for this calculation (e.g. see the (Zhang et al., 2015) you already cite). If you really want to deduce a freshwater concentration, I think you really must try to present it also with an estimate of the (high) uncertainty.

**R: Our assumption is that the subglacial endmember is a mixture of mostly basal ice meltwater and [some] drainage of surface melt to the base of the glacier**

through moulins (Tuckett et al. 2019), although we think that refreezing occurs and limits surface pond drainage to the bedrock. An endmember of 875 nM (±231 nM propagated uncertainty of MWf and plume concentration) is scaled according to the meltwater fraction within the plume. Also, due to the steep topography along the WAP, the residence time of water within the subglacial hydrological system might be shorter than under the larger glaciers elsewhere in Antarctica, which contain subglacial lakes. We can not test this assumption and instead rely on a rough estimate of dFe by scaling to the MWf of the plume.

535 I'm not sure this is surprising, if you look at any studies (either field or lab-based) looking at dFe behavior, you invariably see strong removal at salinities even fractionally above zero (<1) practically immediately (within minutes), so I think it would be correct to say all available data suggests a universal trend in dFe removal on this scale.

**R: We have changed the sentence to indicate intense dFe removal is occurring on this scale, though we mainly refer to a strong oxygen gradient.**

537-540 I'm not sure there is presently evidence to support this, either that dFe concentrations change with glacier type/scale. I haven't looked at this in detail, and this is hard to deduce as there's obviously lots to think about in terms of what concentrations to compare and other confounding factors. In terms of the plume, I think the concentrations here are very similar to those reported for much larger discharge Greenland catchments e.g. (Hopwood et al., 2016; Kanna et al., 2020).

**R: Agreed. There are many factors one could think would have an important control of dFe content, including the bedrock source material, availability of weathering reactants/organic matter, oxygen levels, and residence time in the subglacial hydrologic system. We can remove this sentence (line 537) as the controls on endmember subglacial dissolved trace metal concentrations are unknown.**

I also find this a little confusing (it is clearer after reading the next few paragraphs) as it reads as if the (Death et al., 2014) study is quoting a value of 3-30 uM for the plume, whereas I think this actually refers to zero salinity. I agree, that unless a model manages to formulate the rapid scavenging/removal occurring on very small scales particularly well -most models simply can't do that on this scale because this is subgrid for another other than a regional model- that these values are too high to do what they are being designed to do, but I think the phrasing here could be clearer.

R: We will change this section by adding a sentence about the importance of, and difficulty with parameterizing scavenging/removal at the ice-ocean interface as all studies suggest intense removal of dFe on short time/length scales.

586 are upper limit

**R: Resolved.**

598 Be more specific with what your oxidation rate is referring to, dissolved Fe(II) and dissolved Mn(II)?

**R: We are referring to the oxidation rate of dissolved Fe(II) and Mn(II). We've changed the text to reflect oxidation state of dissolved metals.**

600 I recently read another pre-print concerning Mn and Fe trends in a similar environment which you may find interesting (https://www.essoar.org/doi/10.1002/essoar.10506252.1)

**R: Thank you for bringing this pre-print to our attention.**

Figure 8 How many glacial ice samples are you plotting here? Is there enough data to do this robustly?

R: Fig. 8 displays the average LpMn:LpFe ratio of two glacial ice samples with varying particle loads. Despite the large range in particulate mass embedded, the ratio is approximately the same (0.061±0.002). Therefore, we use this ratio as representative for glacial ice.

620 Is this an increase considering the uncertainty on the values?

R: This is not a significant increase, but instead shows a remarkable consistency between seasons. We do not have a formulated interpretation of why this might be, however, one thought is that this indicates something about the particles present (monodispersive in size? a single source of lithogenics as glacial flour?), and have reached saturation for adsorptive binding of dFe.

626 This seems speculative "are the target for ligand-mediated mineral dissolution and perhaps microbial uptake"

R: We concede that it is speculative, however it is not well known if inorganic colloidal/particulate iron is bioavailable. The presence of strong Fe-binding ligands suggests concentrations of the most bioavailable source Fe' are too low to support optimal growth, and therefore additional Fe is required.

639 Is there a specific reason for a comparison to the California Current transition zone? I don't this discussion adds much, yes there is a huge excess of NO3 pretty much

everywhere across the region, and from a ratio perspective, much of this NO3 will remain following complete dFe drawdown (which is confirmed by time series at bases in the region showing NO3 very rarely approaches low concentrations) – but microbes are still experiencing a high dFe concentration throughout much of the year, so I don't think it's the case that they are Fe-limited in term of their growth during the growth season (or did I misunderstand something here?)

**R: Studies in the CA Current region have established that ratios of NO3:dFe are generally predictive of diatom Fe stress, more so than the dFe concentration. Admittedly, this is a different environment with higher overall dFe concentrations, but the possibility of iron stress on certain populations cannot a priori be excluded.**

654 Raiswell, correct reference? This statement is perhaps is a little too specific, you could comment that the detail of ligand concentration/binding strength is not explicitly represented in most models.

**R: Raiswell reference is removed. This sentence is changed to reflect the lack of accurate ligand representation in biogeochemical models.**

656 I think the earlier (Lippiatt et al., 2010) work argues this.

**R**: We have added the citation here.**

659 "associated feedbacks on climate" this is a big step

**R: Removed and changed to "biogeochemical cycles of the macronutrients."**

678 Are there fjords with strong katabatic winds in the Amundsen Sea?

**R: Katabatic winds are generally present everywhere, and can be associated with moving sea ice and the formation of polynyas. The Amundsen does not have fjords since ice shelves are prominent in this sector.**

686-890 Values like this derived from a hypothetical meltwater endmember need to be flagged as 'rough' or have some uncertainty quantified.

R: Agreed. We will include uncertainties on these estimates, although this section is to illustrate simply that glacial ice meltwater (with a relatively high mean dFe content) cannot account for all dFe within the surface, when biological removal processes are greatly reduced. 693 – I moved this comment having written it earlier in the text – how do you know the meltwater fractions you calculate are all associated with meltwater from this fjord? Presumably it is not, on its own, the major source of meltwater to the region, so other sources, likely some outside your model boxes, are producing meltwater which is then laterally transferred through your region? I think this caveat needs to be explained as at present in many places the text reads as if your fjord was the major source of meltwater (and thus dFe) to the region.

R: This caveat is explained in the limitations of the surface meltwater dye experiments. Through analogous meltwater dye experiments, we checked that glaciers in surrounding bays and Gerlache Strait contribute only 0.0003 MWf, and is presently explained in line 711 – 714. We have changed the sentence, "Third, only meltwater from the inner Andvord Bay is tracked..." to "Third, only meltwater originating from the inner Andvord Bay is considered."

696 "or, alternatively, that the glacial end member concentration is too low" Not sure I see the logic here, only if meltwater had to be 100% of the dFe supply? But we know, as shown in the text, there are multiple sources, so this doesn't make sense

**R: Sentence is revised.**

705-715 This presumably supports the earlier caveat about where meltwater comes from, that the Bay studied is not a/the sole major meltwater source, so the meltwater observed in/around the Bay is coming from multiple places not captured in the model set up?

**R: A potentially small fraction, indicated in line 711.**

785 "leading to enhanced productivity and sedimentation of carbon" You don't show this herein.

**R: Sentence is revised.**

Conclusion – This is quite long and I think would be sharper if cut. The new calculations are interesting but might sit better in the main text.

R: Yes, we will move the calculations to a final section summarizing the detailed modelling components. The conclusion will summarize the main sections of the paper, aided through the visualization, and will delineate outstanding questions not addressed by this work.

852 cause melting (warm-based). (?)

**R: Changed to "warm-water."**

889 I'm not sure you can make a conservative estimate of dFe from TdFe, is there a simple relationship between the two? I would say a 'rough' estimate of 10%.

**R: There is no simple relationship between the two, so the language here is changed to reflect this is a rough estimate.**

References refered to:

Ardiningsih, I., Zhu, K., Lodeiro, P., Gledhill, M., Reichart, G.-J., Achterberg, E. P., Middag, R. and Gerringa, L. J. A.: Iron Speciation in Fram Strait and Over the Northeast Greenland Shelf: An Inter-Comparison Study of Voltammetric Methods , Front. Mar. Sci. , 7, 1203 [online] Available from: https://www.frontiersin.org/article/10.3389/fmars.2020.609379, 2021.

Death, R., Wadham, J. L., Monteiro, F., Le Brocq, A. M., Tranter, M., Ridgwell, A., Dutkiewicz, S. and Raiswell, R.: Antarctic ice sheet fertilises the Southern Ocean, Biogeosciences, 11(10), 2635–2643, doi:10.5194/bg-11-2635-2014, 2014.

Henson, S., Le Moigne, F. and Giering, S.: Drivers of Carbon Export Efficiency in the Global Ocean, Global Biogeochem. Cycles, 33(7), 891–903, doi:https://doi.org/10.1029/2018GB006158, 2019.

Holding, J. M., Markager, S., Juul-Pedersen, T., Paulsen, M. L., Møller, E. F., Meire, L. and Sejr, M. K.: Seasonal and spatial patterns of primary production in a high-latitude fjord affected by Greenland Ice Sheet run-off, Biogeosciences, doi:10.5194/bg-16-3777-2019, 2019.

Hopwood, M. J., Connelly, D. P., Arendt, K. E., Juul-Pedersen, T., Stinchcombe, M. C., Meire, L., Esposito, M. and Krishna, R.: Seasonal Changes in Fe along a Glaciated Greenlandic Fjord, Front. Earth Sci., 4, doi:10.3389/feart.2016.00015, 2016.

Kanna, N., Sugiyama, S., Fukamachi, Y., Nomura, D. and Nishioka, J.: Iron supply by subglacial discharge into a fjord near the front of a marine-terminating glacier in northwestern Greenland, Global Biogeochem. Cycles, doi:10.1029/2020GB006567, 2020.

Lippiatt, S. M., Lohan, M. C. and Bruland, K. W.: The distribution of reactive iron in northern Gulf of Alaska coastal waters, Mar. Chem., 121(1–4), 187–199, doi:10.1016/j.marchem.2010.04.007, 2010.

Menzel Barraqueta, J.-L., Schlosser, C., Planquette, H., Gourain, A., Cheize, M., Boutorh, J., Shelley, R., Pereira Contreira, L., Gledhill, M., Hopwood, M. J., Lherminier, P., Sarthou, G. and Achterberg, E. P.: Aluminium in the North Atlantic Ocean and the Labrador Sea (GEOTRACES GA01 section): roles of continental inputs and biogenic particle removal, Biogeosciences, 2018, 1–28, doi:10.5194/bg-2018-39, 2018.

Zhang, R., John, S. G., Zhang, J., Ren, J., Wu, Y., Zhu, Z., Liu, S., Zhu, X., Marsay, C. M. and Wenger, F.: Transport and reaction of iron and iron stable isotopes in glacial meltwaters on Svalbard near Kongsfjorden: From rivers to estuary to ocean, Earth Planet. Sci. Lett., 424, 201–211, doi:10.1016/j.epsl.2015.05.031, 2015.

**Citations not included in main text, referred to in comment responses:**

Cutter, G. A. and Bruland, K. W.: Rapid and noncontaminating sampling system for trace elements in global ocean surveys, Limnol. Oceanogr. Methods, 10(6), 425–436, doi:https://doi.org/10.4319/lom.2012.10.425, 2012.

Jordan, T. A., Riley, T. R. and Siddoway, C. S.: The geological history and evolution of West Antarctica, Nat. Rev. Earth Environ., 1(2), 117–133, doi:10.1038/s43017-019-0013-6, 2020.

Krisch, S., Hopwood, M. J., Schaffer, J., Al-Hashem, A., Höfer, J., Rutgers van der Loeff, M. M., Conway, T. M., Summers, B. A., Lodeiro, P., Ardiningsih, I., Steffens, T. and Achterberg, E. P.: The 79°N Glacier cavity modulates subglacial iron export to the NE Greenland Shelf, Nat. Commun., 12(1), 3030, doi:10.1038/s41467-021-23093-0, 2021.

Tuckett, P. A., Ely, J. C., Sole, A. J., Livingstone, S. J., Davison, B. J., Melchior van Wessem, J. and Howard, J.: Rapid accelerations of Antarctic Peninsula outlet glaciers driven by surface melt, Nat. Commun., 10(1), 4311, doi:10.1038/s41467-019-12039-2, 2019.

---

## Author Comment (AC2)

Review of 'Seasonal dispersal of fjord meltwaters as an important source of iron to coastal Antarctic phytoplankton' by Forsch et al.

This manuscript aims to constrain the input and dispersal of Fe and Mn rich fjord water in an Antarctic Fjord. This is an interesting and important objective given that these metals are important drivers of primary productivity in the Southern Ocean, including coastal areas. Whereas it is known glacial melt must supply these metals, much is unknown about the underlying processes or the effective fluxes into Fe and or Mn limited waters that are generally located further away from direct sources. To tackle this difficult question, a whole suite of methods is used, including water column profiles, sediment and ice sampling as well as modeling work. This is a very comprehensive study and I really liked to conceptual model (Figure 10) that brings together all the different aspects, however, this clarity and synthesis is somewhat lacking in the text. Despite being generally well written from a language point of view, the text is often not clear (see specific comments) and the text sections are not always well connected. For example, the section on ligands (4.5) provides some discussion on (changes in) ligand concentrations and binding strengths, but ends with a very general section on the potential role of ligands in keeping Fe in solution but no real novel insights from the current data (or comparison to recent insights from an Arctic glacier Fe-speciation study, see specific comments). More importantly, the role of ligands or the balance between solubilization and scavenging is not at all considered in the modelling approach in the next section. In fact, I struggled to see what we learn about Fe in the modelling approach using conservative tracers, given that the elusive balance between solubilization and rapid scavenging / precipitation reactions is one of the most important reasons for our limited understanding of the Fe biogeochemical cycling. In the appendix, a 10% dissolution of TDFe is considered conservative, but no consideration is given to how much of this actually remains in solution and hence is subject to long range transport to Fe-poor regions. I realize it is no easy feat to constrain this, but I think it should be discussed and the modelling section could be shortened considerably.

Overall, I think this manuscript will provide a valuable addition to the literature. Nevertheless, it would benefit from shortening and conveying the novel insights from the data more clearly as well as tying the different sections in the discussion better together (so that it is more one story rather than different aspects that only in the conceptual model really come together). I also noted that while Mn is often mentioned in the manuscript, it does not appear at all in the conclusions whereas I'm confident there are some interesting new insights into the biogeochemical cycle from Mn based on this dataset.

**R: We thank Reviewer #2 for their constructive comments and criticisms. We have made improvements to the discussion of ligands by a comparison to recent literature in the Arctic. Our conservative tracer approach has been informative in terms of parsing out the potential importance of different sources of iron around Antarctica (e.g. Dinniman et al. 2020, St-Laurent et al., 2017). A more complex model which includes iron chemistry is beyond the scope of this work, which is a first look at the potential importance of fjord-based glacial melt as a source of iron to coastal Antarctica. We have also improved our discussion of manganese and place this work in the context of recent literature indicating the role of both Fe and Mn in limiting primary production in the Southern Ocean. We have improved the length of the article by shortening and moving much of the modeling discussion to the supplemental. Additionally, the conclusion is shortened. We address each comment by line using boldface formatted text.**

Specific comments:

In the introduction I missed Mn considering that Mn is known to be (co-)limiting, even in coastal waters such as the Ross Sea (https://doi.org/10.1038/s41467-019-11426-z)

**R: We will make changes to include Mn in our discussion and conclusion. This will be reflected now in the title, discussion of trace element dispersal/implications for downstream primary production, and conclusions.**

120-130 bit late now, but surprised by the choice for stainless steel rather than titanium as used in TM ice core drilling and notably the razor could have easily been replaced by a ceramic knife

**R: Sampling of glacial ice was opportunistic, and precautions were taken to reduce the chance of contamination. We agree that ceramic is a better choice and will be strongly considered for future work.**

292 the work by Bown et al 2017 (https://doi.org/10.1016/j.dsr2.2016.07.004) seems relevant here

**R: Relevant citation added.**

327 not sure I would describe an $R^2$ of 0.48 as highly correlated. And given the equation LpFe = TDFe – dFe, does a correlation between dFe and LpFe indeed imply exchange? I would assume the correlation is inherent to the definition.

**R: We changed this sentence to "correlated." The correlation does not necessarily imply exchange, therefore we have removed this statement.**

389 'particulate Fe are associated with more crystalline and thus less labile Fe oxides'; less labile than the 'comparable fraction is refractory and is not liberated by any of the solution treatments (31%).' mentioned in the previous sentence?

**R: The crystalline Fe oxides mentioned in 388 are more labile than the refractory silicates. These oxides are less labile than the poorly crystalline oxides mentioned in 384. For clarity, the sentence is revised to, "Other fractions of particulate Fe are associated with more crystalline Fe oxides (goethite, hematite) and the minerals magnetite and pyrite."**

475 'Since glacial meltwater is restricted to the surface, it constitutes a significant input of Fe to the surface throughout the growth season'; seems the statement on significance should come after the discussion in the following lines.

**R: This statement will be moved to the end of the paragraph.**

504 what is the statement on light limitation based on?

**R: This is based on our conceptual phenology of the bloom (Pan et al. 2020). We believe for most of the year, phytoplankton growth is light-limited due to low availability of light (steep topography surrounding the fjord blocks sunlight or sea ice cover occurs late-Fall until early Spring). We have added the Pan et al. reference to this sentence.**

517-520 how were crustal vs authigenic material and the reported fraction identified?

**R: These were determined using the following equation, for which average crustal values were taken from estimates of the average composition of the upper continental crust.**

**% crustal = ([TpAl]\*Me:Al$_{crustal}$)/[TpMe]**

**The remaining fraction, after accounting for % biological (0% for all samples), is assumed to be authigenic. This will be clarified in the subsequent version of the manuscript.**

524 'indicating a large oxide fraction is associated with this particulate matter' this statement is not explained (or referenced)

**R: An oxide fraction in plume particulate matter is inferred by comparing TDAl:TDTi (64 mol:mol) and TpAl:TpTi (39 mol:mol). Both ratios are elevated above average upper crust ratios, and the observation that total dissolvable fraction is**

**more enriched in Al than total particulates indicates Al is associated with oxide fractions (Kryc et al. 2003), since Al is a heavily scavenged element by oxyhydroxides at oceanic pH levels and these will solubilize with a dilute HCl leach. We will include a brief explanation in support of this claim.**

534-535 875 nM is higher than TDFe measured (346.95±160.40 nM); why does this suggest settling loss through flocculation is likely occurring? TDFe is also made up of particles that could be emitted from the glacier. I agree there is loss and settling of Fe, but the argument based on TDFe eludes me.

**R: We assume that most of the LpFe is produced during rapid oxidation of the reduced pool of dFe (as Fe(II)aq). Therefore, LpFe+dFe would represent the endmember concentration of Fe(II)aq within zero salinity anoxic subglacial meltwater.**

537 – 539 also, the extrapolation approach here excludes any precipitation and is likely an underestimation of the endmember concentration.

**R: We account for precipitation through the conversion of dFe to LpFe, since LpFe represents a recently precipitated/adsorbed fraction of dFe. See above comment.**

541- 543 'However, subglacial… shelf (Schodlok et al., 2016).' What is the relevance of this sentence, not clear

**R: Advective transport under ice shelves could further reduce the flux of dissolved iron once upwelled into the euphotic zone 10s-100s km away from the point source of meltwater discharge (Krisch et al. 2021). The text was changed to clearly state the relevance.**

543 what export efficiency, previous sentence was on the width of the shelf?

**R: The point here was that boundary scavenging, if not accurately represented, could lead to a large overestimate of the contribution of subglacial meltwaters to the euphotic zone, which is hundreds of km from the point of discharge. We will revise this sentence for clarity.**

558-587 what is the point of this section? The calculated sediment efflux of Fe is much higher than the global average which is somehow related to bioturbation, but how is not made clear. It is stated bioturbation would decrease the efflux, so how does it explain the higher than average efflux? Reference is made to Marsay, but it is not clear to me what the relation is between the current and those observations. This section should be clarified and can probably be shorter/ more succinct

**R: The point of this section is to arrive at a plausible scenario to explain two things: high estimated efflux values based on diffusive flux, and the enrichment of oxides at the sediment-water interface.**

**DFe maxima down-core could support high concentrations of dFe in surficial porewaters. We believe that the high rates of bioturbation and abundance of organic matter following bloom sedimentation could sustain dissimilatory processes deeper in the cores, which results in down-core local maxima in dFe. Diffusion coefficients based on bioturbation are smaller than those based on molecular diffusion, therefore we do not know if bioturbation enhances the dFe flux from the sediments. However, we do think that bioturbation enhances redox cycling within the sediments through the mixing process. The enrichment of labile Fe oxides in the top few centimeters demonstrates rapid oxidation of reduced Fe, and potential formation of colloidal Fe and labile particulate Fe.**

**Additionally, we point out that the global average from Dale et al. (referenced line 568) is for continental margins, so one would expect fluxes in the fjord to be considerably different since they are coastal.**

**Efflux estimates in Marsay et al. (2014) might more accurately show the effect of these processes on the efflux from reducing sediments to the water column following significant oxidation and reversible scavenging, since we argue diffusive flux estimates are upper limits (line 585). Therefore, large uncertainties remain as to the net flux of dFe from sediments following rapid oxidation at the sediment-water interface. The diffusion coefficients of colloidal Fe have not been determined, but would be substantially lower (by orders of magnitude) compared to Fe(II).**

**We agree with the reviewer that this section can be considerably improved and condensed.**

561 'The result is deviation from results based on diffusion alone' Rephrase

**R: Since diffusion coefficients based on bioturbation are smaller than those based on molecular diffusion, we have removed this sentence as we do not have evidence to support the claim that epibenthic fauna affect the sedimentary efflux of metals. However, as noted in the previous response, redox cycling is enhanced through bioturbation. We will reference Burdige and Christensen (submitted to Gochem. Cosmochim. Acta).**

564 greater deposition than what?

**Compared to data from the adjacent continental shelf. Sentence will be revised.**

577 what flux estimates, those of Pb?

**R: We are referring to dFe flux estimates. This will be explicitly stated.**

582 DIR is never used

**R: Removed.**

621 how does a scaling of dFe to LpFe correspond to an increase in eL between seasons. Please clarify

**R: Excess ligands, defined in line 616, also scale with LpFe. However, we can change our choice of word ("correspond") since there is no clear reason for why an excess of ligands would scale with increased dFe and LpFe.**

625 how does this compare to Arctic glacier work? Could ligands be outcompeted by particle scavenging? (Ardiningsih et al., 2020 https://doi.org/10.1016/j.marchem.2020.103815)

**R: It is possible that particle surface sites are responsible for the large excess of ligands in the fjord, especially in the Fall. However, our samples are filtered of particles >0.4 μm, thus we would not observe this. In the Fall, a greater $L_t$:dFe ratio compared to late-Spring, combined with a lower average conditional stability constant of the ligand pool results in a lower complexation capacity and inferred ability to compete with particle binding sites. However, we might then expect for the fraction of dFe and LpFe to reflect a greater enrichment of particles, which is not observed.**

**A sentence to this effect (with citation) will be added to the revised manuscript.**

636-637 so the excess (strong?) ligands originate from Bransfield Strait? Not clear

**R: It is possible that the ligands originate from outside the fjord, or there is a source common to these coastal regions. We will add text as to the possible sources of the ligands considering recent literature. For example, exopolymeric substances (EPS) are a well-defined pool of strong ligands present at high concentrations in sea ice (Lannuzel et al. 2015). Following sea ice melt, these ligands, presumably in far excess of dFe in sea ice, could be released into the surface waters. A short discussion on the [potential] sources of ligands will be included in the subsequent version.**

642-643 Not necessarily, only if ligand induced dissolution occurs which is not demonstrated here

**R: The sentence is made more speculative.**

650-660 some general global implications regarding ligands are mentioned, but what is the insight generated from the current data?

**R: We will keep the first couple of sentences here, and then focus our discussion to this study. We believe the most important finding is the seasonal change in quality and quantity of ligands. In the bloom initiation, overall ligand strengths are higher than in the Fall. However, concentrations of ligands increase following the bloom. Concurrently, dFe concentrations increase and do not saturate the ligands to the same extent as in late-Spring. This is due to a lower complexation capacity of the ligand pool resulting from a considerably weaker ligand pool. Therefore, the dFe pool in the Fall is more subject to boundary scavenging (as free Fe).**

663 what is the relation between icebergs, vertical shear and katabatic winds? Mentioning of icebergs seems out of place here

**R: For clarity, mention of icebergs is removed. The line of thought is that icebergs would provide additional shear and thus, mixing with katabatic winds and "stir" the upper water column.**

665 what is meant with 'export the surface layer efficiently'

**R: 'Efficiently' is removed.**

679 export of surface water?

**R: Yes, surface water. Resolved.**

682-697 what is the point of this exercise were Fe is assumed to behave conservatively (i.e. like $\delta^{18}O$ used to estimate the meltwater fraction) whereas we know it does not, especially not in a productive region. The final statement (it is expected that the input of glacial meltwater throughout the melt season would supply some dFe to the surface) could have been made without this section.

**R: The high concentrations of organic ligands found within the fjord offer a possible means of stabilizing a significant fraction of dFe. The modeling study provided insight on the distribution of dFe throughout the fjord region and thus**

local variations in dFe availability for primary producers. However, we will leave the final statement and take this opportunity to reduce the text length since a comparison to the Gerlache Strait is not needed when the results are too variable.

698-723 this whole section does not contribute a single insight into the biogeochemical cycle of Fe (or new insights into the actual dispersion of melt water), so what does it contribute to the ms?

R: We included this section to highlight the limitations of using the dye experiments to model the dispersion of surface glacial meltwater. This text will be condensed and moved to the supplementary as we think it is important to consider. The main text will mention that the limitation of using the dye experiments are discussed in supplementary under a heading to that effect.

700 the $\delta^{18}O$ approach should differentiate between meteoric and sea ice melt so the MWf should not be influenced by sea ice melt.

R: The modeled meltwater flux is different than the flux inferred by isotopic mass balance. The issue with the mass balance estimate is that export of meltwater implies non-steady state conditions and will result in an underestimate of meteoric input to the fjord. Pictures from the glacial cameras displayed large amounts of icebergs lingering in the fjord over long periods, which is a likely source of meteoric meltwater that cannot be modelled and was attempted to be parameterized with the modelled meltwater flux. The modeled flux "best recreates observed salinity and temperature profiles in Andvord Bay…over 4 months (Hahn-Woernle et al. 2020)." (line 703)

724-729 this section I found very confusing; dFe observations were made prior to a wind event; the model predicts upwelling if there is wind and the model results of upwelling are supported by the observations prior to upwelling? Also, how am I to see the elevated dFe in late spring at station S3 in fig3? No stations are labelled and the color scale is more or less homogenous (all blue).

R: Station S3 has been added to fig. 3. Multiple re-occupations of S3 during the late Spring cruise (LMG15010) showed elevated surface dFe concentrations following the Dec. 10 wind event. We refer to the supplemental for dFe concentrations to demonstrate an increase in concentration in the surface during the late Spring cruise: [dFe]$_{avg}$ (prior to Dec. 10) = 3.19 nM; [dFe]$_{avg}$ (after Dec. 10) = 4.14 nM (compared to average surface value within the fjord = 2.47 nM;

**line 289). We use dFe observations prior to upwelling with the model-derived vertical velocities to derive the upwelling Fe flux.**

**The color scale will be modified to highlight the variability in [dFe] in the late Spring.**

746 Only if Fe remains in solution and is not taken up by phytoplankton along the way, do we know anything about this?

**R: The length-scale for dFe in productive surface waters is typically very short (e-fold length-scale ~25 km.). Thus, with some distance from the continental shelf, much of the dFe may be removed if not stabilized as nanoparticles or organic ligands. Annett et al. (2017), show for the '600 line' on the PAL LTER sampling grid, concentrations of dFe decrease from ~0.9 nM to 0.1 nM within 30 km of the coast. The authors suggest organic ligands are responsible for stabilizing a consistent 0.1 nM dFe over the entire continental shelf. However, we believe that since sampling occurred during peak bloom conditions, that biological removal is the primary control of concentrations in this location.**

**From our study, we note that Station B (Fig. 1) located on the continental shelf has a surface dFe concentration of ~1.2 nM in the Fall. This instead could represent the late-productive season concentration remaining in surface waters following replenishment from coastal sources. This could be confirmed by measuring the organic ligands from these shelf waters.**

756 this point was just made in line 734

**R: We make the distinction between vertical mixing (diapycnal) and localized upwelling (displacement of subsurface waters). Both processes are consequences of the katabatic winds investigated in the dye experiments.**

768 what geochemical data?

**R: This statement has been revised. At present, the Mn:Fe geochemical data does not allow us to distinguish sedimentary versus subglacial sources since distinct oxidation kinetics of Fe and Mn will alter these endmember ratios towards higher values.**

774-775 confusing sentence; export of subsurface dye leads to a low surface concentration due to proximity to the surface?

**R: Upper ocean physics is controlled by the katabatic winds, which exports the surface layer across the fjord mouth. Thus, subsurface water entrained into the surface are also subject to export resulting in low concentrations within the fjord. We will re-phrase.**

775 'The upper ocean is more subject to changes in the upper ocean dynamics' that seems like an open door to me...

**R: Changed to "subject to changes in wind stress."**

783 not sure it was shown, more argued / suggested

**R: Changed to "argued."**

786-787 how does the comparison of dFe and the meltwater fraction in fig 3 'support to the modeled dynamics' ?

**R: Elevated concentrations of dFe and meltwater are found within the inner fjord and at Sill 3. Both of these locations are identified as places where upwelling of subsurface water masses, potentially containing subglacial plume water, occurs. This explanation is added to the text.**

807 why would intensifying coastal currents lessen the likelihood for pulsed export of meltwater-derived Fe?

**R: Coastal currents develop fronts that prevent lateral exchange. The strength of these fronts is dependent on the densities of coastal currents relative to the meltwater-influenced water masses.**

826 where is Barilari Bay and why is it relevant?

**R: Barilari Bay is a fjord containing tidewater glaciers influenced by warm (above-freezing) ocean temperatures, yet is similarly productive to Andvord Bay. The location of Barilari Bay is now briefly described in the text.**

828-829 how can a single wind event along the WAP export 36 km$^3$ of meltwater if the total Antarctic meltwater discharge is only 32.5 – 97.5 km$^3$ yr$^{-1}$?

**R: The focus of the Pattyn et al. (2010) is on modeling basal melt production due to geothermal heating. This estimate does not consider basal melting of ice shelves in contact with warm ocean water and is therefore an inaccurate (low)**

**estimate for total meltwater. Since Pattyn et al. is unrealistic, we will compare more recent, improved estimations of meltwater flux.**

833 also larger than the total annual meltwater discharge. Something is incorrect or very unclear

**R: See previous comment.**

843 why would a phytoplankton bloom lead to decreased ligand concentrations, not clear and contradictive to section 4.5

**R: Since increased turbidity is expected to decrease the magnitude of local phytoplankton blooms, the source of ligands decreases. The sentence is re-phrased: "To a first approximation, decreases in the magnitude of local phytoplankton blooms and associated ligand sources is expected to reduce efficacy of solubilization of particulate Fe and natural fertilization downstream resulting from this leaky fjord."**

**Citations not included in main text, referred to in comment responses:**

Kryc, K. A., Murray, R. W. and Murray, D. W.: Al-to-oxide and Ti-to-organic linkages in biogenic sediment: relationships to paleo-export production and bulk Al/Ti, Earth Planet. Sci. Lett., 211(1), 125–141, doi:https://doi.org/10.1016/S0012-821X(03)00136-5, 2003.

Dinniman, M. S., St-Laurent, P., Arrigo, K. R., Hofmann, E. E. and van Dijken, G. L.: Analysis of Iron Sources in Antarctic Continental Shelf Waters, J. Geophys. Res. Ocean., 125(5), e2019JC015736, doi:https://doi.org/10.1029/2019JC015736, 2020.

Krisch, S., Hopwood, M. J., Schaffer, J., Al-Hashem, A., Höfer, J., Rutgers van der Loeff, M. M., Conway, T. M., Summers, B. A., Lodeiro, P., Ardiningsih, I., Steffens, T. and Achterberg, E. P.: The 79°N Glacier cavity modulates subglacial iron export to the NE Greenland Shelf, Nat. Commun., 12(1), 3030, doi:10.1038/s41467-021-23093-0, 2021.

Burdige, D., and Christensen, J.: Iron biogeochemistry in sediments on the western continental shelf of the Antarctic Peninsual. (submitted to Geochem. Cosmochim. Acta).

Lannuzel, D., Grotti, M., Abelmoschi, M. L. and van der Merwe, P.: Organic ligands control the concentrations of dissolved iron in Antarctic sea ice, Mar. Chem., 174, 120–130, doi:https://doi.org/10.1016/j.marchem.2015.05.005, 2015.

---

## Referee Report (RR1)

Seasonal dispersal of fjord meltwaters as an important source of iron and manganese to coastal Antarctic phytoplankton

I reviewed the original submission of this article. The authors have appropriately discussed and replied to two reviewers, both of whom found the original article interesting. I have no major comments on the revised version or on the authors' replies. The manuscript is technically sound and a substantial, and very useful, study.

Minor technical edits suggested below:

30 primary production increases and potentially carbon export -suggest 'primary production, and potentially carbon export, increase'

33 Delete 'existing'

39 ACC – defined? You don't use it much, suggest just writing it out

115 'In brief…' Doesn't read well. I suggest you just move this sentence before the sampling, 'Bottles were pre-cleaned…' and then the paragraph will read ok

142 0.34 nM dFe

144-150 No units on any concentration

193 Buck et al., ref format

202 were previously applied

313 Suggest splitting this sentence in two

323 Fig. 5 n6 ??

501 I agree generally, but you do not explicitly show Fe or Mn is limiting herein, suggest 'could stimulate'

623 More convincing is the correlation between LpFe and dFe so suggest just referring to this directly. I suspect the 4% and 5% have relatively large standard deviations so wouldn't state it was 'remarkable'.

801 small fraction of Fe from these particles

821 I understand this long term perspective, but potentially important to stress this is long term given the earlier comment on the potential for short-term increases, so maybe add a clause 'In the short term, increased Fe and or Mn supply may increase primary production…. But conversely, in the long term….'

Supplement

40 mwf of

---

## Author Response (AR2)

**Response to Reviewer #1 on behalf of all coauthors**

 Seasonal dispersal of fjord meltwaters as an important source of iron and manganese to coastal Antarctic phytoplankton
I reviewed the original submission of this article. The authors have appropriately discussed and replied to two reviewers, both of whom found the original article interesting. I have no major comments on the revised version or on the authors' _replies. The manuscript is technically sound and a substantial, and very useful, study.

**We thank Reviewer #1 again for helpful comments and recommended edits. These improvements make the message from this work clearer, precise, and compelling. Responses to reviewer comments are provided below in-line.**

Minor technical edits suggested below:

30 primary production increases and potentially carbon export -suggest 'primary production, and potentially carbon export, increase' _

**We have changed accordingly.**

33 Delete 'existing' _

**Deleted.**

39 ACC – _defined? You don't use it much, suggest just writing it out

**Antarctic Circumpolar Current is now written.**

115 'In brief…' _Doesn't read well. I suggest you just move this sentence before the sampling, 'Bottles were pre-cleaned…' _and then the paragraph will read ok

**Moved.**

142 0.34 nM dFe

**Specified, dFe.**

144-150 No units on any concentration

**Units are added.**

193 Buck et al., ref format

**Reference is fixed.**

202 were previously applied

**Fixed.**

313 (**318**) Suggest splitting this sentence in two

**Fixed.**

323 Fig. 5 n6 ??

**Fixed.**

501 I agree generally, but you do not explicitly show Fe or Mn is limiting herein, suggest 'could stimulate' _

**Changed.**

623 (**628)** More convincing is the correlation between LpFe and dFe so suggest just referring to this directly. I suspect the 4% and 5% have relatively large standard deviations so wouldn't state it was 'remarkable'.

**Comparison to LpFe is now stated. We have removed 'remarkable.'**

**They do have large standard deviations: 4±3% late Spring; 5±2% Fall.**

801 small fraction of Fe from these particles

**Fixed.**

821 I understand this long term perspective, but potentially important to stress this is long term given the earlier comment on the potential for short-term increases, so maybe add a clause 'In the short term, increased Fe and or Mn supply may increase primary production…. But conversely, in the long term….' _

**We've added this clause for distinguishing between short term responses to climate change, and longer term changes with changes in ice volume.**

Supplement
40 mwf of

**Fixed.**

**Response to Reviewer #2 on behalf of all coauthors**

The authors have done a good job revising the manuscript and I only have a view remaining comments.

**We thank Reviewer #2 for an additional round of careful review of the manuscript. The suggestions made here will improve the precision of the language and the overall clarity for readers, which is always of the greatest importance. We reply to each comment in-line.**

125-133 I fully understand and respect the explanation that the ice sampling was opportunistic. However, stating 'Acute attention to cleanliness was applied….' does (at least to me) imply the authors were following best practice rather than doing the best with the tools available. To avoid setting a precedent for future research by others, please state explicitly the recommended tools (no steel tools, ceramic knive, etc).

**Absolutely, we agree with this. A sentence of caution is added and a recommendation for alternatives is provided.**

191 not my area of expertise, but a recent publication in this journal (doi.org/10.5194/bg-2021-134) indicates the used method 'causes overestimation of [L] and could result in a false distinction into more than one ligand group'. The current ms is not a methods paper, but maybe the implications of such bias should be discussed in this ms?

**While overestimation is a potential issue with using these methods, the trends within the dataset, including the interpretations are sound including comparing to existing datasets using these methods. The Gerringa et al. (2021) reference applied these methods to model ligands, and not environmental samples. We have included a sentence to this effect in section 3.5 and the reference provided by the reviewer.**

345 how was the plume defined (i.e. what where the criteria for 'in the plume' vs 'out of the plume')?

**The plume was defined in lines 273 – 275: "subsurface neutrally-buoyant plume (~100m) characterized by a point source of relatively cold and particle-laden water emanating from the terminus of Bagshawe Glacier…".**

485 I indeed suggested to move the sentence, but moving it to the current location has not improved the clarity of this section; at least I do not follow how the discussion concerning a negative relation with MWf and metal ratios in particles, leads to the conclusion meltwater is a significant source of Fe throughout the growth season.

**Sentence removed for section clarity. In the late Spring, a negative relationship occurred because of intense removal of surface reservoir (winter reservoir) due to phytoplankton uptake and stratification of the surface (prevents surface replenishment). The point that meltwater is a significant source is solely based on the very high concentrations of terrigenous trace metals in meltwater (relative to seawater) and the supply, which is input directly to the euphotic zone.**

508 is such a buoyant plume here observed? Was mentioned in the results but would be good to stress here again

**Now explicitly stated.**

525 'Cold-water glaciers are locations where the subglacial environment flows directly into the fjord with minimal mixing with seawater.' seems contradictive to line 510 stating that cold tongues 'entrain deep water masses'. could you clarify?

**Clarified. Low meltwater production rates do not induce extensive overturning and turbulence. Both are essential for vertical currents and ambient seawater entrainment.**

535-546 the sentences in this section seem somewhat disjointed (i.e. hard for a non-specialist to follow the narrative/rationale).
**We have added some clarifying phrases to the text.**

548-566 what is the connection between flocculation (1st paragraph), resuspension (2nd paragraph) and diffusion-based sediment fluxes (3rd paragraph), not very clear to me.

**This section was focused on the role of sediments as a source to the water column, and that particulate material is derived from the subglacial plumes.**

567 what are these results, the flux estimates? the 12 µmol m-2 d-1 is ~factor 4 lower than the estimate in the previous paragraph, is it implied these are similar or relatively different?

**Clarified. We imply that the results are not surprising for a coastal setting with high dissolved oxygen because our estimated fluxes are higher than, but similar in magnitude to, the average for continental shelves.**

633 I meant Ardiningsih et al., 2020 https://doi.org/10.1016/j.marchem.2020.103815) as a comparison to Arctic work

**Yes, the wrong citation was listed. We've corrected this, although the 2021 (*Biogeosci.*) publication does have some relevance to this work.**

**We have added a sentence comparing the findings of these works in Antarctica and the Arctic: large source of undersaturated weaker ligands close to glaciers compared to adjacent open ocean stations.**

661 sounds similar to interpretations for an Arctic system by Ardiningsih et al., 2020 (direct citation from abstract 'The lower ligand binding strength in the outflow results in a higher inorganic Fe concentration, [Fe´], which is more prone to precipitation and/or scavenging than Fe complexed with stronger ligands.'). Seems noteworthy to me to discuss such similarities.

And how does this section relate to the previous section 'In the Fall, ….. fraction of dFe and TDFe does not reflect a greater enrichment of particles.' Seems contradictive; first lower complexation capacity is not deemed to lead to partitioning to particles (i.e. scavenging) whereas here it is stated the DFe pool might be more subject to scavenging. Please clarify.

**The dFe pool may be more subject to scavenging in the Fall, but we do not see this in the particulate Fe speciation data (near constant ratio of dFe:LpFe). We now make the point that the greater excess of weaker ligands in the Fall is due to a large source of these ligands and they are not saturated because there is a smaller increase in dFe concentrations. This contrasts with competition with particles, which might remove bound dFe from ligand complexes, but we don't see this.**

662 Ardiningsih et al., 2020 can be updated to the published paper (rather than discussion paper)
671 export the surface layer sounds odd to me, do you mean it leads to mixing of surface and deeper waters (i.e. vertical export) or that fjord surface water advects offshore (i.e. horizontal export)?

**Clarified by saying, "can laterally export the surface layer".**

675-676 deepening of the mixed layer by 25 m and water upwells from 150 m depth. Can you clarify what happens where?

**Water that is being upwelled would displace surface water by moving it laterally, whereas winds also add energy internal to a water parcel by increasing turbulence, thus increasing the depth of well-mixed surface water. Upwelling was found to occur at both Sill 3 and in the inner basin A. These are also locations where mixing increased, indicated by a deeper**

**mixed layer. Both of these processes had varying strengths at either location (Sill 3 and IBA).**

690 what stations are referred to with 'both stations'?

**Sill 3 and Gerlache Strait stations. Fixed.**

703-708 I commented on this last time, and the response file gives an explanation, but the changes indicated in the response do not seem to be made in the text (same figure, no explanation the station was occupied before and after the wind event with different observed concentrations). Moreover, based on the supplemental table I do not get the average value of 4.14 after December 11 for the shallowest sampled depth at station S3, and given the range in values, I highly doubt the concentration difference is significant.

**The explanation is now included in the text, although you are correct, the difference is not significant despite there being a higher concentration at S3 following the wind event (3.94 nM not 4.14 nM, miscalculated). Late Spring average 3.19±1.53 nM dFe; Fall average 3.94±1.92 nM dFe.**

730 previous section was also (partially) on upwelling at station S3 which is not the glacier terminus so I suggest some rearranging/clarifying of the text as it is not always obvious which process is discussed in which section.

**We have added some text to the manuscript to improve clarity.**

748 quicker than what?

**The deep dye. Clarified.**

774-787 I like this section, but think you should at least mention that the actual 'effective' fluxes of Mn and Fe will be much lower due to non-conservative behavior of these elements.

**We have added a sentence about the uncertainty of the effective fluxes due to scavenging and biological uptake during transport offshore.**

---

## Author Response (AR3)

Dear Associate Editor,

On behalf of all coauthors, we have made the technical corrections indicated in your report:

Meltwater fraction is introduced prior to using the acronym (MWf).

We changed the reference for estimating diffusive flux from sediments.

When possible, figures do not use red and green colors, simultaneously. For example, Figure 4 was fixed.

We appreciate the handling of our manuscript through this process, and it has been our pleasure to work with you, the editorial team, and the referees.

Thank you,
Kiefer and co-authors